

# Quantifying the effect of mixing on the mean Age of Air in CCMVal-2 and CCMI-1 models

Dietmüller Simone[1], Eichinger Roland[2,1], Garny Hella[1,2], Birner Thomas[3], Boenisch Harald[4], Pitari Giovanni[5], Mancini Eva[6], Visioni Daniele[6], Stenke Andrea[7], Revell Laura[8], Rozanov Eugene[7,9], Plummer David A.[10], Scinocca John[11], Jöckel Patrick[1], Oman Luke[12], Deushi Makoto[13], Kiyotaka Shibata[14], Kinnison Douglas E.[15], Garcia Rolando[15], Morgenstern Olaf[16], Zeng Guang[16], Stone Kane Adam[17,18], and Schofield Robyn[17,18]

[1]Deutsches Zentrum für Luft- und Raumfahrt (DLR), Institut für Physik der Atmosphäre, Oberpfaffenhofen, Germany
[2]Ludwig Maximilians Universität, Institut für Meteorologie, Munich, Germany
[3]Colorado State University, Fort Collins, Colorado, USA; currently sabbatical leave at DLR
[4]Karlsruhe Institute of Technology (KIT), Germany
[5]Department of Physical and Chemical Sciences, Università dell'Aquila, L'Aquila, Italy
[6]Department of Physical and Chemical Sciences and center of Excellence CETEMPS, Università dell'Aquila, L'Aquila, Italy
[7]Institute for Atmospheric and Climate Science, ETH Zürich (ETHZ), Zürich, Switzerland
[8]Bodeker Scientific, Christchurch, New Zealand
[9]Physical-Meteorological Observatory/World Radiation Center, Davos, Switzerland
[10]Environment and Climate Change Canada, Climate Research Division, Montréal, QC, Canada
[11]Environment and Climate Change Canada, Climate Research Division, Victoria, BC, Canada
[12]National Aeronautics and Space Administration Goddard Space Flight Center (NASA GSFC), Greenbelt, Maryland, USA
[13]Meteorological Research Institute (MRI), Tsukuba, Japan
[14]School of Environmental Science and Engineering, Kochi University of Technology, Kami, Japan
[15]National Center for Atmospheric Research (NCAR), Boulder, Colorado, USA
[16]National Institute of Water and Atmospheric Research (NIWA), Wellington, New Zealand
[17]School of Earth Sciences, University of Melbourne, Melbourne, Australia
[18]ARC Centre of Excellence for Climate System Science, Sydney, Australia

*Correspondence to:* S. Dietmüller (simone.dietmueller@dlr.de)

**Abstract.** Stratospheric age of air (AoA) is a useful measure of the overall capabilities of a general circulation model (GCM) to simulate stratospheric transport. Previous studies have reported a large spread in the simulation of AoA by GCMs and coupled chemistry-climate models (CCMs). Compared to observational estimates simulated AoA is mostly too low. Here we attempt to untangle the processes that lead to the AoA differences between the models and between models and observations.

5 AoA is influenced by both, mean transport along the residual circulation and two-way mixing; we quantify the effects of these processes using data from the CCM inter-comparison projects CCMVal-2 and CCMI-1. Transport along the residual circulation is measured by the residual circulation transit time (RCTT). We interpret the difference between AoA and RCTT as additional aging by mixing. Aging by mixing thus includes mixing on both the resolved and subgrid scale. We find that the spread in AoA between the models is primarily caused by differences in the effects of mixing, and only to some extent by differences in

10 residual circulation strength. These effects are quantified by the mixing efficiency, a measure of the relative increase of AoA by mixing. The mixing efficiency varies strongly between the models from 0.21 to 0.99. We show that the mixing efficiency is not



only controlled by horizontal mixing, but by vertical mixing and vertical diffusion as well. Possible causes for the differences in the models' mixing efficiencies are discussed. Differences in subgrid scale mixing (including differences in advection schemes and model resolutions) likely contribute to the differences in mixing efficiency. However, differences in the relative contribution of resolved versus parametrized wave forcing do not appear to be related to differences in mixing efficiency or AoA.

## 1  Introduction

The Brewer-Dobson-circulation (BDC) affects the stratospheric distribution of radiative active trace gases, which strongly contribute to the radiative forcing of the climate system. Stratospheric mean age of air (AoA) is a measure of the strength of transport along the BDC and is defined as the mean transport time of an air parcel from the entry region at the tropical tropopause to any region in the stratosphere (Hall and Plumb, 1994; Waugh and Hall, 2002). AoA is a useful measure for the analysis of stratospheric transport, as it includes both the effects of the slow overturning residual circulation and the effect of the two-way mass exchange of air parcels, referred to as (eddy) mixing (e.g. Butchart, 2014). AoA can be also derived from observations of conserved tracers whose tropospheric concentrations increase approximately linearly over time, such as balloon-borne and satellite observations of $SF_6$ or $CO_2$ mixing ratios (e.g. Andrews et al., 2001; Engel et al., 2009, 2017; Stiller et al., 2012; Haenel et al., 2015). AoA derived from observations then can be directly compared to AoA simulated by GCMs and CCMs (as done e.g. in Eyring et al., 2006; SPARC CCMVal, 2010). The concept of stratospheric AoA is very helpful, as it is the only possible observation-based measure of the BDC. However, it is important to note that the AoA diagnostic bears information on both mean residual circulation and effects of two-way mixing, as it is the integrated effect of all transport processes.

In the past model-inter-comparison studies with GCMs, chemical transport models (CTMs) and CCMs (e.g. Hall et al., 1999; Eyring et al., 2006; Butchart et al., 2010) showed a significant model spread in AoA. In comparison to observations, simulated AoA was too low in many models, mainly in the middle and upper stratosphere. The model-inter-comparison activity CCMVal-2 (Chemistry-Climate Model Validation activity 2) was conducted with the goal to improve the understanding of stratosphere-resolving CCMs. In the SPARC (Stratospheric Processes and their Role in Climate Project) CCMVal-2 report (SPARC CCMVal, 2010) the AoA diagnostics of 15 CCMVal-2 models were analyzed at a wide range of latitudes and altitudes. The models AoA was compared to in-situ observations of Andrews et al. (2001) and Engel et al. (2009) (see the tropical and mid-latitude AoA profiles and the latitudinal AoA distribution at 50 hPa in figure 5.5 of SPARC CCMVal (2010)). It was shown that 7 out of 15 models match closely the observed AoA at 50 hPa at all latitudes and also their vertical tropical AoA profiles are within the uncertainties of the observations at all altitudes. However, for most of these models AoA is too low in the middle stratosphere when compared to in-situ observations. Moreover, the spread of simulated AoA between the models is high.

To understand the model spread in AoA and their discrepancy to observations, the processes that drive stratospheric transport need to be disentangled, namely the effects of residual transport and mixing. Several methods for this separation have been



used. Ray et al. (2010) have used a methodology based on the conceptual one-dimensional tropical leaky pipe (TLP) model (see Neu and Plumb, 1999) to constrain the circulation strength and mixing strength across the subtropical barrier in a model by observed concentrations of long-lived tracers. In SPARC CCMVal (2010) several diagnostics were employed to measure transport characteristics like tropical ascent or tropical to mid-latitude mixing. Those diagnostics were based on tracer concen-

trations, allowing for a comparison to observations. However, with most diagnostics it is not possible to entirely separate the different effects. It was found that most models appear to have too strong tropical-to-mid-latitude mixing and too fast tropical ascent. As those two biases compensate, it is argued that despite those model biases a reasonable AoA can be produced in the models. Overall, a good relationship between the model's ability to simulate mean AoA to the ability to simulate both, tropical lower stratospheric ascent, and tropical-mid-latitude mixing was found (see SPARC CCMVal (2010), their Fig. 5.20).

Recent studies have developed diagnostics to quantify the effect of mixing on AoA from model data (e.g. Garny et al., 2014; Ploeger et al., 2015b; Dietmüller et al., 2017). Garny et al. (2014) quantified the effect of mixing on AoA (termed as aging by mixing) with the global climate model ECHAM6 (European Centre/Hamburg version 6). They analyzed the difference of simulated AoA and the transit time of the hypothetical transport along the residual circulation only (in the following termed as

residual circulation transit time, RCTT). They found that additional aging by mixing can be found in most of the stratosphere, because mixing between the tropics and extratropics causes air to recirculate, and thus AoA is increased. Only in the lowermost stratosphere, where air mass exchange with young tropospheric air occurs, mixing leads to a reduction of AoA. Ploeger et al. (2015b) confirmed these results with the Lagrangian chemistry transport model CLaMS (Chemical Lagrangian Model of the Stratosphere) by explicitly calculating aging by mixing on resolved scales through integration of local eddy mixing tendencies

along the residual circulation trajectories. In the explicit calculation of aging by mixing, parametrized and numerical diffusion are not included. Dietmüller et al. (2017) combined the two methods of calculating aging by mixing and thus the effects of resolved and unresolved mixing on AoA (latter termed aging by diffusion) can be separated. By analyzing simulation data of the CCM EMAC (ECHAM/MESSy Atmospheric Chemistry) and of CLaMS, they found that aging by diffusion enhances AoA, contradicting some previous thoughts, which assumed that diffusion makes air younger (e.g. Eluszkiewicz et al., 2000;

Waugh and Hall, 2002; SPARC CCMVal, 2010). However, the contribution of unresolved mixing was found to only play a minor role (impact of 5-10% on AoA) in both models.

By applying the concept of the idealized TLP model, Garny et al. (2014) derived the so-called 'mixing efficiency'. The mixing efficiency is defined as the ratio of the two-way mixing mass flux across the subtropical barrier to the net residual mass flux. This mixing efficiency controls the ratio of tropical mean AoA to RCTT, and thus describes the relative increase in AoA

by mixing. Garny et al. (2014) investigated the mixing efficiency for three different climate equilibrium states (pre-industrial (1860), present-day (1990), and future (2050)) and found that the strength of two-way mixing is tightly coupled to the strength of the lower stratospheric residual circulation. The ratio of mixing strength to residual circulation strength is almost constant in the three different climate states (i.e. the mixing efficiency is constant). Garny et al. (2014) proposed that the comparison of the relative aging by mixing (or of the mixing efficiency) between models can provide useful insights in the widely known





model deficits in the AoA simulation.

In this study we seek to gain a better quantitative understanding of the processes that control the BDC, in order to explain the differences in climatological AoA between CCMs. To do so the effects of residual transport and of mixing on AoA are an-
alyzed and investigated for various recent CCMs. We use the data of the hind-cast simulations of the inter-comparison projects CCMVal-2 and CCMI-1 (Chemistry Climate Model Initiative, phase 1). A brief description of the models and simulations is presented in Section 2. The methods of calculating AoA, RCTT, aging by mixing, the mixing efficiency and tropical upwelling are shortly introduced in Section 3. Annual mean AoA, RCTT, aging by mixing and mixing efficiency are analyzed in Section 4. In Section 5 we discuss possible causes for the inter-model differences in mixing, including effects of vertical dispersion
and model characteristics. A summary and concluding remarks are given in Section 6.

## 2  CCM simulations analyzed in this study

In the present study, we analyze the model output from 17 state-of-the-art CCM simulations. The output of 8 simulations is obtained from the coordinated model inter-comparison Chemistry-Climate Model Validation activity 2 (CCMVal-2, Morgenstern et al. (2010); Eyring et al. (2006)) and the output of the other 9 simulations from the ongoing Chemistry Climate Model Initia-
tive phase 1 (CCMI-1, Eyring et al. (2013); Morgenstern et al. (2017)). A list of these CCMs is provided in Table 1, together with references and relevant information on the model setups, namely the vertical and horizontal resolution, the height of the model top, and the advection scheme. This sub-set of models that contributed to CCMVal-2 and CCMI-1 is chosen according to the availability of the necessary data (AoA and residual circulation velocities).

In the following, we briefly describe some aspects of the CCMs that are relevant for our study. A detailed overview of all models that participated in CCMVal-2 and CCMI-1 is provided by Morgenstern et al. (2010, 2017). Note that many CCMI-1 models have a predecessor model in CCMVal-2, thus the development since CCMVal-2 (e.g. improvements in chemistry and physics or higher resolution) can be studied. Note also that there are family relationships between different models, e.g. the models ACCESS-CCM and NIWA-UKCA are identical and the models EMAC and SOCOL are both based on the ECHAM5
climate model. Moreover, we use the EMAC model in two different vertical resolutions (i.e. EMAC-L47 and EMAC-L90MA). The models' horizontal resolutions vary between ∼5°and ∼2°and the vertical resolutions range from 26 to 126 levels with the top of the different models between 0.07 hPa up to 0.00005 hPa.
Several types of advection schemes are used in the CCMVal-2 and CCMI-1 models. Numerical diffusion in GCMs is linked to the discrete nature of grids which are used for transport processes. Generally, advection schemes are designed to minimize
numerical diffusion, however, for stability reasons several models require explicit diffusion (Morgenstern et al., 2010, 2017). The different advection schemes are also provided in Table 1. Note that in CCMVal-2 there are two models (MRI and SOCOL) that use different schemes for meteorological and chemical tracers. Thus, in these models, the advection of the different types of tracers is physically not self-consistent (Morgenstern et al., 2010). The SOCOL model has changed the advection scheme



between CCMVal-2 and CCMI-1. Differences in the advection scheme may cause differences in the distribution of chemical species and AoA, particularly in the lower stratosphere (Morgenstern et al., 2010; Eluszkiewicz et al., 2000).

The BDC is driven by the momentum deposition of breaking waves (Haynes et al., 1991) with small-scale gravity waves contributing significantly, but these small-scale waves are not resolved in most GCMs. Numerous parametrization schemes for the

5   calculation of gravity wave drag (GWD) are applied in the different CCMVal-2 and CCMI-1 models. Based on the generation of the gravity wave scheme, the computation of their drag is separated into orographic and non-orographic parameterization schemes. For the non-orographic gravity wave drag, various methods are used to determine the sources as well as the launch levels of the gravity waves. Basically, each model uses another combination of these schemes, an overview is provided in Table S9 of Morgenstern et al. (2017) and in Table 3 of Morgenstern et al. (2010).

The simulations evaluated here are the long transient (free running) reference simulations REF-B1 of CCMVal-2 (covering the recent past from 1960-2006) and REF-C1 of CCMI-1 (covering the recent past from 1960-2010). The long-term mean over those years provides the base for our inter-comparison. The REF-B1 and REF-C1 reference simulations were performed analogously, using observational forcings, including all anthropogenic and natural forcings based on changes in trace gases, solar

15   variability, volcanic eruptions and sea surface temperatures. Some of these forcings, however, differ between CCMVal-2 and CCMI-1. All details of the REF-B1 and REF-C1 simulations are documented by Morgenstern et al. (2010, 2017) and follow the designs of Eyring et al. (2006) or Eyring et al. (2013).



**Table 1.** Overview of the CCMs and their simulation set-ups used in the present study. The reference(s), the horizontal and vertical resolution (number of model layers), the model top and the advection schemes (of chemical tracers) of the individual models of CCMVal-2 (upper rows) and CCMI-1 (lower rows) are listed. For the spectral models, horizontal resolution is given as triangular truncation of the spectral domain, with T21≈ 5.65°x5.65°, T30/T32≈3.75°x3.75°,T42≈2.8°x2.8°, T47≈2.5°x2.5°. TL159≈ 1.1°x1.1° The advection schemes are SP=spectral, FFSL=flux-form semi-Lagrangian, SL=semi-Lagrangian, STFD=spectral transform and finite difference, FFEE=flux form Eulerian explicit, FV=finite volume (for details see SPARC CCMVal (2010)).

| Model | Reference(s) | Resolution | Top of model | Advection Scheme |
|---|---|---|---|---|
| **CCMVal-2** | | | | |
| CMAM | Scinocca et al. (2008) | T31, L71 | 0.00081hPa | SP |
| GEOSCCM | Pawson et al. (2008) | 2.0°x2.5°, L72 | 0.015hPa | FFSL |
| LMDZrepro | Jourdain et al. (2008) | 2.5°x3.8°, L50 | 0.07hPa | FV |
| MRI | Shibata and Deushi (2008) | T42, L68 | 0.01hPa | STFD* |
| SOCOL | Schraner et al. (2008) | T30, L39 | 0.01 hPa | SL* |
| ULAQ | Pitari et al. (2002) | 11.5°x22.5°, L26 | 0.04 hPa | FFEE |
| UMUKCA-METO | Morgenstern et al. (2009) | 2.5°x3.8°, L60 | 84km | SL |
| WACCM | Garcia et al. (2007) | 1.9°x2.5°, L66 | 0.00005hPa | FFSL |
| **CCMI-1** | | | | |
| ACCESS-CCM | Morgenstern et al. (2009, 2013); Stone et al. (2015) | 2.5°x3.8°, L60 | 84km | SL |
| CMAM | Jonsson et al. (2004) Scinocca et al. (2008) | T47, L71 | 0.0008hPa | SP |
| CESM1-WACCM | Solomon et al. (2015); Garcia et al. (2017) Marsh et al. (2013) | 1.9°x2.5°, L66 | 140km | FFSL |
| EMAC-L90 | Jöckel et al. (2010, 2016) | T42, L90MA | 0.01hPa | FFSL |
| EMAC-L47 | Jöckel et al. (2010, 2016) | T42, L47 | 0.01hPa | FFSL |
| GEOSCCM | Molod et al. (2012, 2015) Oman et al. (2011, 2013) | 2°x2.5°, L72 | 0.015hPa | FFSL |
| MRI | Deushi and Shibata (2011) Yukimoto et al. (2011, 2012) | TL159, L80 | 0.01 hPa | SL |
| SOCOL | Stenke et al. (2013); Revell et al. (2015) | T42, L39 | 0.01 hPa | FFSL |
| NIWA-UKCA | Morgenstern et al. (2009, 2013) Stone et al. (2015) | 2.5°x3.8°, L60 | 84km | SL |
| ULAQ | Pitari et al. (2014) | T21, L126 | 0.04hPa | FFEE |

*these models use different transport schemes for meteorological tracers



## 3   Methods

### 3.1   Calculation of AoA, RCTT and aging by mixing

Stratospheric mean age of air is defined as the residence time of air parcels in the stratosphere, starting at the tropical tropopause (e.g Hall and Plumb, 1994; Waugh and Hall, 2002) and is affected both by the residual circulation, and by eddy mixing. In

almost all CCMs, the AoA tracer is implemented as an inert tracer with prescribed lower boundary conditions ( in some models the lower boundary condition is applied globally in others only in the tropics) that linearly increase in mixing ratio over time ("clock-tracer"; Hall and Plumb (1994)). Diagnosed AoA at a certain grid point in the stratosphere is then calculated as the time lag between the local tracer mixing ratio (at this certain grid point) and the current mixing ratio at a reference point. As this reference point does vary among the models (e.g. boundary layer, tropical thermal tropopause, 100hPa), we subtract the AoA

value at each model's individual tropical thermal tropopause from AoA (so that AoA=0 there), to obtain consistency between the models. Only the CCMVal-2 model CMAM uses a stratospheric source AoA tracer (for details see SPARC CCMVal, 2010)

Karlsruhe, Germany

The residual circulation transit time (RCTT) is the hypothetical age air would have if it was only transported by the residual

circulation, i.e. without eddy mixing. RCTTs are calculated following Birner and Bönisch (2011) by calculating backward trajectories that are driven by the Transformed Eulerian Mean (TEM) meridional and vertical monthly velocities (referred to as residual velocities) with a standard fourth-order Runge-Kutta integration. The backward trajectories are initialized on a latitude pressure grid (depending on the model). The residual meridional velocity $\bar{v}^*$ and the vertical velocity $\bar{w}^*$ are available in the CCMI-1 and CCMVal-2 data base. The backward trajectories are terminated when they reach the thermal tropopause. The

elapsed time is then the residual circulation transit time. A detailed description is given by Birner and Bönisch (2011) and by Garny et al. (2014).

Besides the transport through the residual circulation, AoA is affected by eddy mixing (Neu and Plumb, 1999; Garny et al., 2014; Ploeger et al., 2015b, a). As pointed out by Garny et al. (2014), mixing of air from the mid-latitudes into the tropical pipe

can cause additional aging through recirculation of aged air. This process is called aging by mixing. In their study, Garny et al. (2014) proposed that in global models aging by mixing can be interpreted as difference between simulated AoA and RCTT. However, it has to be taken into account that aging by mixing obtained as difference between AoA and RCTT includes mixing on unresolved scales (namely parameterized and numerical diffusion).

### 30   3.2   TLP Model and mixing efficiency

We use the concept of the Topical Leaky Pipe (TLP) model (Neu and Plumb, 1999) to better understand the contribution of different processes to AoA. The TLP model is a simple one-dimensional conceptual model of stratospheric transport, which includes advection and horizontal two-way mixing between tropics and extratropics across the subtropical barrier. When ne-





glecting vertical diffusion, an analytical solution for tropical and mid-latitude AoA can be formulated. The two free parameters that AoA depends on are the advective vertical velocity (i.e. the residual velocity) and the amount of mixing between the tropics and extratropics, controlled by the so-called mixing efficiency $\epsilon$. The mixing efficiency is defined as the ratio of the mixing mass flux to the net mass flux across the subtropical barrier. At the same time the mixing efficiency is proportional to

the relative increase of AoA by mixing, and it proved to be a useful measure of the relative mixing effects (see Garny et al., 2014). The mixing efficiency can be calculated from model data given the tropical profile of AoA and the vertical residual velocity $\overline{w}^*$. The tropical profiles are averaged over the latitudinal band of $20°$S-$20°$N (sensitivity to the width of the tropical band is discussed in Section 4) and are interpolated to vertical coordinates relative to the tropopause height of each model. The mixing efficiency is then obtained by a best fit for the altitude range from the tropopause to $32\,\mathrm{km}$ (details for the calculation

of the mixing efficiency are given in Garny et al., 2014)

To analyze the role of vertical diffusion for AoA profiles and the derived mixing efficiency (see Section 5.1) the TLP model is implemented as Lagrangian model (following Ray et al. (2014)). Briefly, the model consists of three vertical "pipes" (tropics, northern hemisphere (NH) and southern hemisphere (SH)), and particles are injected in the tropics, advected vertically

with given vertical winds, and exchanged between tropics and the NH and SH extratropics. Horizontal advection and mixing is modeled as "Bernoulli process" based on probabilities of parcel exchange. Vertical diffusion (which is neglected in the analytical TLP solution) is implemented as random walk: The height of each parcel $i$ is calculated as $z_i(t+\delta t) = z_i(t) + \zeta$, where $\zeta$ is a random displacement drawn from a Gaussian distribution with zero mean and variance $\sigma^2 = 2K\delta t$ (where $K$ is the vertical diffusivity at this height; see Ghoniem and Sherman (1985)).

### 3.3 Tropical upwelling

The stratospheric circulation is driven by the dissipation of waves that propagate upwards from the troposphere to the stratosphere. As measure for the strength of the residual circulation, the strength of tropical upwelling is commonly used (Holton et al., 1995). Here, we use the quasi-geostrophic approximation of the transformed Eulerian-mean (TEM) equations to calculate the streamfunction of the residual circulation $\overline{\chi}^*$ driven by the Eliassen-Palm flux divergence (EPFD) and the sum of

orographic and non-orographic gravity wave drag (OGWD and NOGWD) as follows:

$$\overline{\chi}^*_{\overline{m}_0}(p) = \int\limits_p^0 \left[ \frac{1}{cos(\phi) \cdot f} \left( \frac{1}{r \cdot cos\phi} \nabla \cdot F - \frac{\partial \overline{u}}{\partial t} + \overline{X} \right) \right]_{\phi = \phi(\overline{m}_0)} . \tag{1}$$

Here, $F$ denotes the Eliassen-Palm flux, $\overline{X}$ the total zonal gravity wave drag, $f$ the coriolis parameter, $\phi$ the given latitude, p the given pressure and $\overline{m} = r \cdot cos(\phi)(\overline{u} + r \cdot \Omega cos(\phi))$ the meridional gradient of the zonal mean angular momentum. Tropical upwelling is then given by the difference in the residual streamfunction at the tropical boundaries ($20°$S, $20°$N). This calculation

allows to linearly separate the influence of resolved planetary wave driving (EPFD: $\frac{1}{r \cdot cos\phi} \nabla \cdot F - \frac{\partial \overline{u}}{\partial t}$) and unresolved gravity wave drag (GWD: $\overline{X}$) on tropical upwelling. This can provide insights into the driving mechanisms of stratospheric transport and mixing variations, and thereby in AoA spread among the models.



## 4 Effects of mixing on AoA in analyzed CCMs

### 4.1 AoA, RCTT and aging by mixing

The long-term climatological mean AoA, RCTT, and aging by mixing are calculated for each model listed in Table 1 and are shown in Figure 1 for the CCMVal-2 models and in Figure 2 for the CCMI-1 models. Additionally, the residual circulation

is overlaid in the RCTT panels. The climatological means are calculated over the years 1980 to 2006 for CCMVal-2 REF-B1 models and from 1980 to 2010 for CCMI REF-C1 models, because all available simulations overlap in this period. In general, the zonal annual mean patterns of AoA of all CCMs (Fig. 1 and Fig. 2, left panel), agree qualitatively in the typical AoA distribution. All models have lower AoA in the tropical lower stratosphere and old air in the extra-tropical middle stratosphere. However, the simulated magnitude of AoA shows large variations among the different models of CCMVal-2 and CCMI-1,

mainly at high latitudes in the upper stratosphere. In this region, the AoA values range between 4.0 and 6.5 years. Generally, the highest AoA values are found in the UMUKCA-METO (CCMVal-2), lying far outside of the model spread. For the CCMVal-2 models (Fig. 1), besides the UMUKCA-METO model, the ULAQ and MRI models simulated rather high AoA values and the SOCOL model has the lowest AoA values. Within the CCMI-1 models, EMAC and MRI are on the high side of AoA values, whereas the models NIWA-UKCA, SOCOL, ULAQ and WACCM are on the low side. Furthermore, differences

in the shape of the AoA isopleths between the analyzed CCMs are apparent, ranging from peaked to flat gradients. Figure 1 and 2 show strong horizontal gradients for the models GEOSCCM and UMUKCA-METO of CCMVal-2 and for the model MRI of CCMI-1 and low gradients for the model SOCOL of CCMVal-2 and for the models NIWA-UKCA, SOCOL and ULAQ of CCMI-1. Note that the CCMs NIWA-UMUKCA and ACCESS are identical and use the same model-setup for the REF-C1 simulations, however they were conducted on two different platforms. We found that the two model runs are climatological

identical for dynamics (as seen e.g. for upwelling, residual circulation and zonal winds) and also for transport-determined tracers (e.g. $CH_4$). However, there are significant differences in AoA between the two models (with considerably lower AoA in ACCESS), that we can currently not explain. If the platform dependence was the reason for differences in transport, we would expect that similar differences are found in other tracers. Therefore we will only show the results of NIWA-UMUKCA in the following.

For a more quantitative comparison, we show (analogously to chapter 5 of SPARC CCMVal (2010)) the tropical (10°N-10°S) and mid-latitude (35°N - 45°N) annual mean AoA profile and the latitudinal distribution of AoA at 50 hPa for all analyzed CCMs together with the available observed AoA profiles in Fig. 3. The observational data are obtained from airbone in-situ observations of the $SF_6$ and $CO_2$ profiles from different measurement campaigns during the last decades (Andrews et al., 2001;

Engel et al., 2009, 2017). For the mid-latitudes we use the AoA profiles of Engel et al. (2009) and for the tropics the AoA profiles of Andrews et al. (2001). The observational uncertainty in AoA for the data of Engel et al. (2009) includes both, trace gas uncertainty and variability of AoA over 30 years (see Engel et al., 2009), whereas the observed tropical AoA profiles of Andrews et al. (2001) were not reported with uncertainties. The latitudinal distribution is compared to the in-situ data of Andrews et al. (2001). As observation-based in-situ measurements of the BDC are sparse, we also use for the latitudinal distribution AoA





values derived from $SF_6$ satellite observations from the ENVISAT Michelson Inferometer for Passive Atmospheric Sounding (MIPAS) (Stiller et al., 2012; Haenel et al., 2015). However, AoA derived from observed $SF_6$ is overestimated because of the mesospheric sinks of $SF_6$ (Haenel et al., 2015; Ray and Andrews, 2017). The uncertainty of observational latitudinal AoA profile is shown as range between maximum and minimum AoA values.

The tropical AoA profile (Fig 3a), which is influenced by the ascent in the tropics, vertical diffusion and horizontal mixing across the subtropics (see SPARC CCMVal, 2010), shows increasing AoA values with altitude. We find that throughout the stratosphere many models have lower AoA values compared to the observations of Andrews et al. (2001), apart from the UMUKCA-METO model whose air is 1-2 years older. Regarding the inter-model differences in the tropical profiles of AoA, we find a large spread between the various models: the standard deviation of the AoA multi-model mean is about 10% at 20

hPa and 30% at 70 hPa (excluding the outlier model UMUKCA-METO). The mid-latitude AoA (Fig 3b) is influenced by the ascent in the tropics, the mixing across the subtropical barrier, the descent in polar regions and by the degree of polar vortex isolation. Its profile is characterized by a rapid AoA increase with altitude in the lower stratosphere and nearly constant AoA values above. Stratospheric air in the CCMVal-2 model UMUKCA-METO is very old (outlier), however compared to in-situ observations (Engel et al., 2009) again AoA in most models is slightly too young. This is mainly the case for the middle and

upper stratosphere, but in the lower stratosphere AoA from many models is within the range of uncertainty. Mid-latitude AoA profiles also show high inter-model spread, with standard deviations of about 15% at 20 hPa and 20% at 70 hPa. In Fig. 3c the simulated AoA (CCMVal-2 and CCMI-1) at 50 hPa at all latitudes is compared to MIPAS observations and to in-situ observations of Andrews et al. (2001). Except for UMUKCA-METO, all models show younger air than observed, particularly at high latitudes. However, especially at high latitudes AoA derived from observed $SF_6$ is overestimated because of the mesospheric

sinks of $SF_6$ (see Haenel et al., 2015; Ray and Andrews, 2017). Overall, we can say that compared to observations, AoA is too low in most of the models analyzed in our study. The fact that AoA in CCMVal-2 models is too low compared to observations has been reported before (see fig. 5.5, in chapter 5 of (SPARC CCMVal, 2010)).

As discussed in the introduction, we separate the effect of transport along the residual circulation (RCTT, Fig.1 and 2, middle panel) and the integrated effect of eddy mixing (aging by mixing, Fig. 1 and 2, right panel) on the simulated AoA. First,

the model differences in the RCTTs are discussed. All CCMs show a quite consistent structure in the RCTTs, with strong meridional gradients mainly in the mid-latitudes and high latitudes. All RCTTs follow the structure of the residual circulation (see overlaid red and blue contours in the RCTT panels). However, inter-model differences in RCTT are apparent. Maximum RCTT values range between about 3 and 5 years in polar regions, with the ULAQ model of CCMI-1 having the lowest transit time (and thus the fastest circulation) and the CMAM model of CCMVal-2 having the highest transit times (and thus slowest

circulation). Regarding the structures of the RCTTs, the models CMAM, GEOSCCM, SOCOL and WACCM from CCMVal-2 and the models CMAM, EMAC and GEOSCCM of CCMI-1 show two minima in the RCTT in the subtropics. In contrast, the remaining CCMs show one wide RCTT minimum in the subtropics. Whether there is one wide minimum or two minima is probably a question of the seasonal cycle of the circulation. The CCMVal-2 model LMDZrepro has additional circulation cells of poleward transport at high latitudes in the residual circulation. This is reflected in the RCTTs by vertical gradients at high





latitdes.

As seen in previous studies (e.g. Garny et al., 2014), AoA significantly differs from RCTT in magnitude and structure (see Fig. 1 and 2). Thus, aging by mixing (interpreted as the difference of AoA and RCTT (see Garny et al., 2014)) plays an impor-

tant role for AoA. Figures 1 and 2 (right panels) consistently show for all models that mixing leads to additional aging of air in most parts of the stratosphere, with maximum values in aging by mixing in the subtropical upper stratosphere. Only in the extratropical lowermost stratosphere, where mixing with tropospheric air occurs, mixing leads to younger air (see minimum aging by mixing values there). Similar structures of aging by mixing are found in all CCMs, but quantitative differences are apparent. Aging by mixing varies between 2.5 and 3.5 years, with the models CMAM, SOCOL of CCMVal-2 and the models

CMAM, SOCOL and NIWA-UKCA of CCMI-1 having the lowest aging by mixing values, and the models UMUKCA-METO and LMDZrepro of CCMVal-2 and EMAC and MRI of CCMI-1 having the highest aging by mixing values. Note that numerical and vertical diffusion is included in that aging by mixing term. Recently, Dietmüller et al. (2017) separated the effects of resolved aging by mixing (by explicitly calculating daily local mixing tendencies along the residual circulation trajectories) and unresolved aging by mixing (referred to as "aging by diffusion") in two global models. Note that one of these models

was EMAC-L90, and we analyze the identical simulation here. They found for both models that numerical diffusion makes air slightly older (aging by diffusion impacts AoA by about 10%). Another conclusion of this study was, that the contribution of aging by diffusion on AoA is different in magnitude and distribution in the two models, mainly because they have different advection schemes. Thus, differences in unresolved mixing likely contributes to inter-model differences in aging by mixing. We discuss this issue in section 5, where we qualitatively relate model characteristics (i.e. advection scheme and resolution) to

AoA.

We also address the question whether simulated AoA (and thus CCM transport) improved in CCMI-1 compared to CCMVal-2. Eyring et al. (2006) analyzed CCMs from CCMVal-1 and reported that AoA in CCMVal-1 models is improved compared to previous model-data inter-comparisons. In SPARC CCMVal (2010) CCMs that participated both in CCMVal-1 and in CCMVal-

2 were compared and no clear improvement in the simulation of AoA could be found. In our study the AoA performance for all analyzed CCMI-1 models that have a predecessor model in CCMVal-2, i.e. the models CMAM, MRI, GEOSCCM, SOCOL, ULAQ and WACCM (see Table 1) are examined (Fig. 3, dashed lines for CCMVal-2 and solid lines for CCMI-1). The AoA model spread is not reduced for the CCMI-1 REF-C1 simulations compared to the CCMVal-2 REF-B1 simulations. Additionally we find that in most CCMI-1 models air is even younger than in their CCMVal-2 predecessor models (except for MRI,

and tropical AoA values of SOCOL), and thus the simulations with the predecessor models agree better with observations. However, some forcings used in the CCMI-1 REF-C1 and the CCMVal-2 REF-B1 simulations are not identical. E.g. one significant difference is the inclusion of an additional major volcanic injection of aerosol into the stratosphere in the CCMI-1 volcanic forcing data-set (see Morgenstern et al., 2017). This could explain the lower AoA in CCMI-1 REF-C1 simulations, as AoA in model simulations tends to be lowered by major volcanic eruptions at higher altitudes (30 hPa), as recently shown

by Pitari et al. (2014). However, this also means that we cannot clearly separate the effect of differences in forcing and model



improvement (e.g. higher resolution in CCMI-1 REF-C1 simulations).





**Figure 1.** Zonal annual mean of AoA (left), RCTT (middle) and aging by mixing (right). Annual means show the average over the years 1980-2006 for the REF-B1 simulations of CCMVal-2. Units are given in years. Annual mean residual circulation is overlaid over the RCTT patterns (blue and red lines).



**Figure 2.** As Fig. 1, but annual means show the average over the years 1980-2010 for the REF-C1 simulations of CCMI-1.





**Figure 3.** (a) Tropical ($10°$N-$10°$S), (b) mid-latitude ($35°$N-$45°$N) AoA profile and (c) latitudinal AoA distribution at 50 hPa for all analyzed CCMVal-2 models (dashed lines) and CCMI-1 models (solid line), with AoA averaged over the years 1980-2006. AoA profiles are shown together with the observational AoA data derived from airbone in-situ measurements of $SF_6$ (black dots) and $CO_2$ (black crosses). For the extra-tropics the observations from Engel et al. (2009) and for the tropics the observations of Andrews et al. (2001) are used. Uncertainties of the observational data of Engel et al. (2009) are shown as $1\sigma$. Observational data of Andrews et al. (2001) were not reported with uncertainties. The latitudinal AoA distribution is shown together with MIPAS data (grey diamond symbols) and in-situ measurements of Andrews et al. (2001) (black cross for $CO_2$ and black dot for $SF_6$). Error bars of the observational data at 50 hPa give the range between minimum and maximum values.




### 4.2 Inter-model correlation of tropical upwelling with RCTT and AoA

The residual circulation is often measured by the strength of tropical upwelling, commonly used at 70 hPa. In the following we investigate whether tropical upwelling is a good measure of the transport times along the residual circulation throughout the stratosphere. We calculate the correlation of climatologies of mean tropical upwelling with the corresponding RCTTs across

all model simulations. Tropical upwelling is averaged between the individual turnaround latitudes of each model, respectively. Fig. 4 shows the correlations between RCTTs and tropical upwelling at 80 hPa, 70 hPa and 50 hPa. All panels in Fig. 4 mostly show negative correlations, which indicates that stronger tropical upwelling leads to reduced transit times through acceleration of the residual circulation. The highest correlations can be found for tropical upwelling at 70 hPa. Here, the correlation reaches values between 0.7 and 0.8. These maxima can be found in the tropical pipe as well as in the downwelling branches of the

BDC in the extratropics. The maximal correlation of tropical upwelling at 50 hPa with the RCTTs is found between 30 and 10 hPa and the structure resembles the deep BDC branch. The correlation with the tropical upwelling at 80 hPa is generally weaker and has its maxima in the lower extratropical stratosphere, i.e. in the region of the shallow branch of the BDC. These results indicates that tropical upwelling is a good measure of transport along the residual circulation, in particular at 70 hPa, while tropical upwelling above relates to transport in the deep branch, and below to the shallow branch of the BDC. Due to the

relatively small sample size of 17 models, these correlations are statistically significant only in a few regions. These, however, do support the basic idea of the points made above.

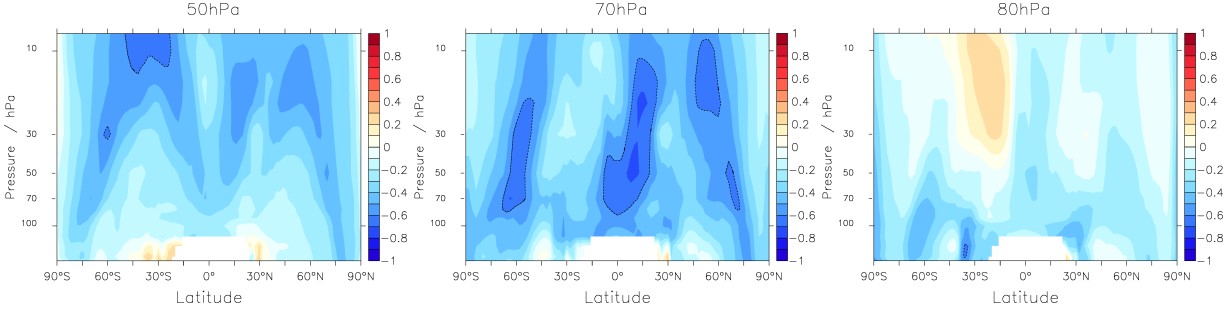

**Figure 4.** Inter-model correlation coefficients for the correlation between RCTTs and tropical upwelling calculated at the turnaround latitudes at 50 hPa (right), at 70 hPa (middle) and at 80 hPa (left). The contoured lines mark the level of significance on the 95% level.

Additionally, in Fig. 5, the correlations of tropical upwelling with AoA are shown. In general, the correlations of tropical upwelling with AoA are far weaker than for the RCTTs and thus also not significant. The patterns seen in Fig. 4 are not visible

here. Again, the highest correlations are found for tropical upwelling at 70 hPa with maxima reaching values around 0.5 in the extratropical lower stratosphere. For tropical upwelling at 50 hPa, hardly any correlation with AoA can be seen and tropical upwelling at 80 hPa only weakly correlates with AoA. As for the RCTTs strongest correlations are found in the extratropical lower stratosphere (only in the NH). Interestingly, in particular in the tropical pipe correlations are lower compared to the extratropics (see 70 hPa tropical upwelling). This indicates that additional processes that act locally on AoA in the tropics play a



role here, as for example tropical vertical diffusion. The low and insignificant correlations of tropical upwelling to AoA among all models show that mixing in general plays an important role for the simulation of AoA, and that its relative effect on AoA is differently strong in different models. A more quantitative contribution of AoA to RCTTs and mixing follows in Section 4.3.

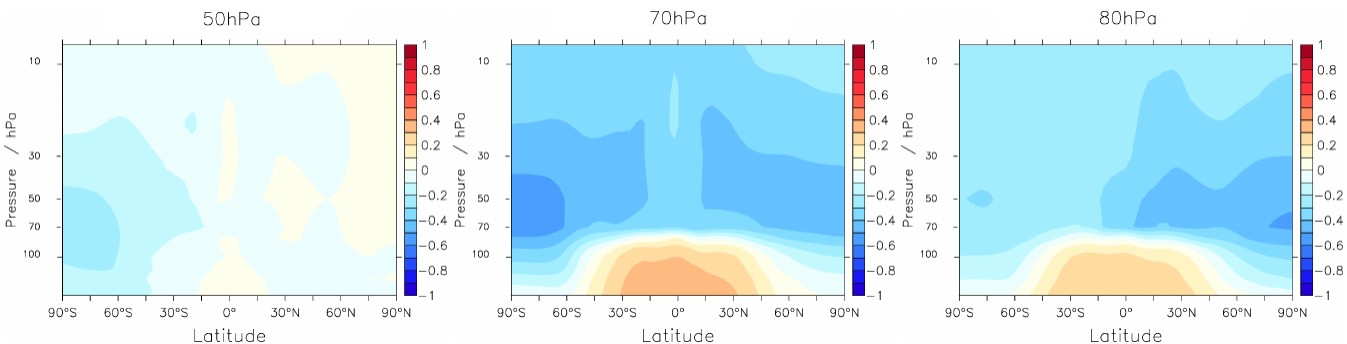

**Figure 5.** As Fig. 4 but correlation between AoA and tropical upwelling. Note that the correlations are not significant on the 95% level anywhere.

## 4.3  Mixing efficiency

In section 4.1, we showed that AoA is influenced by the residual circulation and by mixing. However, these two processes are not independent, as both are linked to wave forcing (e.g. Garny et al., 2014). Furthermore, aging by mixing depends on the speed of recirculation, so that a stronger residual circulation also leads to lower aging by mixing, even if the strength of mixing itself does not change. To get a more independent measure of the mixing strength, we use the mixing efficiency $\epsilon$. This measure is proportional to the relative increase of AoA due to mixing (i.e $\sim (AoA - RCTT)/RCTT$). Thus $\epsilon$=0 refers to no mixing (and AoA=RCTT) and increasing values of $\epsilon$ referring to an increase in relative mixing strength. The original definition of the mixing efficiency stems from the theoretical concept of the TLP model (see section 3.2), where the mixing efficiency is defined as the ratio of the mixing mass flux to the net mass flux across the tropical barrier. However, in this formulation of the TLP model, vertical mixing or diffusion is neglected. Any numerical or parametrized diffusion both horizontally and vertically will influence tracer transport in the global model. The mixing efficiency calculated from the AoA fields of the models should therefore be interpreted as measure of the relative enhancement of AoA by any mixing or diffusion.

Table 2 lists the derived mixing efficiencies for all model simulations. For the individual CCMVal-2 models the mixing efficiency varies between 0.21 (SOCOL) and 0.99 (UMUKCA-METO). Note that the mixing efficiency of UMUKCA-METO lies far outside the typical range of the other models' mixing efficiency (from about 0.21-0.66). The CCMVal-2 multi-model mean of $\epsilon$ is 0.35 with a standard deviation of 38% (see Table 3, first column). Note, however, that for calculating this multi-model mean UMUKCA-METO and ULAQ are excluded (if not: multi-model mean of 0.47 with a standard deviation of 56%). For the CCMI-1 models the spread in mixing efficiency is smaller, ranging from 0.21 (SOCOL) to 0.5 (MRI). The multi-model mean




of the CCMI-1 model mixing efficiency of 0.32 is similar to that of the CCMVal-2 models. The standard deviation of 27%, however, is much smaller (see Table 3, first column). Taking into account all models together (CCMVal-2 and CCMI-1), the mean mixing efficiency is 0.35 with a standard deviation of 39%. Sensitivity experiments with TLP calculations for two different tropical pipe definitions (i.e 30°N-30°S and turnaround latitudes) were conducted. These sensitivity experiments show that the variation in mixing efficiency does not decrease when using the model's individual turnaround latitudes (see Table 3, second and third column). Thus, it can be concluded that the large differences in $\epsilon$ between models cannot be explained by the fact that the various models have different widths of the tropical band.

The large CCMVal-2 and CCMI-1 model spreads in $\epsilon$ indicate that the relative mixing strength (i.e. the amount of any kind of mixing relative to the strength of the residual circulation) varies strongly among the different models, or in other words, mixing leads to differently strong relative enhancements of AoA.

Figure 6 shows the relationship between tropical AoA and tropical RCTT (Fig. 6a) and between tropical AoA and mixing efficiency (Fig. 6b) for all analyzed CCMVal-2 (crosses) and CCMI-1 (dots) models. Tropical values are all averaged over 20°N-20°S and are given at 10 km above the tropopause (corresponding to approximately 20 hPa). As shown in Fig. 6a, tropical AoA is poorly correlated with tropical RCTT. The correlation coefficient for CCMVal-2 models is only 0.25, however it increases to 0.51 when neglecting the outlier model UMUKCA-METO. For the CCMI-1 models, the correlation is only 0.09 (see Fig. 6a), and for all models (CCMVal-2 and CCMI-1) the correlation is 0.24. Thus, the differences in AoA between the models can be explained only to a very small degree by differences in the strength of the residual circulation. In contrast, a high correlation is found between the tropical AoA and the mixing efficiency (Fig. 6b) with a correlation coefficient of 0.86 for CCMVal-2, of 0.94 for CCMI-1 and of 0.88 for all analyzed models. The relation of tropical AoA to RCTT and the mixing efficiency is shown here exemplary for 10 km above the tropopause, but the result of the strong relation of AoA to the mixing efficiency and the weak relation of AoA to RCTT holds for all heights (not shown here). We conclude that the differences in the mixing efficiency between the models can explain large parts of the spread in simulated AoA. For example for the outlier model UMUKCA-METO the very high AoA value can be explained with a very high mixing efficiency of 0.99, while the RCTT of UMUKCA-METO lies in the same range as other models (see Fig. 6a). Thus, it is not a particularly slow circulation that leads to high AoA in UMUKCA-METO, but relatively large mixing. Similarly, models producing the young air (SOCOL REF-B1 and REF-C1) do not have the fastest circulation, but a low mixing efficiency.

Further, we compare CCMI-1 models with their CCMVal-2 predecessor models, to analyze if there is a systematic change with respect to mixing efficiency in the more recent CCMI-1 simulations. Table 2 shows that $\epsilon$ changes from CCMVal-2 to CCMI-1 are very small ($< 3\%$) for GEOSCCM, MRI and minor ($< 15\%$) for CMAM and WACCM. For SOCOL the mixing efficiency does not change at all. The two models with the highest mixing efficiency in CCMVal-2 show a significant change in $\epsilon$ in CCMI-1: in ULAQ, the mixing efficiency decreases from 0.66 to 0.31, and in the UKCA model from 0.99 to 0.33. Thus in both cases the mixing efficiency lies much closer to the multi-model mean in CCMI-1. Reasons for this will be discussed in the Section 5.2.





**Table 2.** Mixing efficiency $\epsilon$ for all CCMVal-2 REF-B1 (left) and CCMI-1 REF-C1 (right) simulations, used in this study. $\epsilon$ is derived from the TLP model, with the border of the tropical pipe ranging between 20°N and 20°S.

| CCMVal-2 | $\epsilon$ | CCMI-1 | $\epsilon$ |
|---|---|---|---|
| CMAM | 0.25 | CMAM | 0.28 |
|  |  | EMAC-L90 | 0.41 |
|  |  | EMAC-L47 | 0.29 |
| GEOSCCM | 0.28 | GEOSCCM | 0.29 |
| LMDZrepro | 0.55 |  |  |
| MRI | 0.49 | MRI | 0.50 |
| SOCOL | 0.21 | SOCOL | 0.21 |
| ULAQ | 0.66 | ULAQ | 0.31 |
| UMUKCA-METO | 0.99 | NIWA-UKCA | 0.33 |
| WACCAM | 0.32 | CESMA-WACCM | 0.27 |

**Table 3.** CCMVal-2 REF-B1 and the CCMI-1 REF-C1 multi-model mean of mixing efficiency $\epsilon$ and its 1 $\sigma$ inter-model standard deviation (in %). $\epsilon$ is derived by the TLP model, using three different tropical pipe definitions: 20°S-20°N, 30°S-30°N, and turnaround latitudes (TR). Note that the multi-model means of $\epsilon$ exclude the models UMUKCA-METO and ULAQ (CCMVal-2).

|  | $\epsilon$(20°S-20°N) | $\epsilon$(30°S-30°N) | $\epsilon$(TR) |
|---|---|---|---|
| CCMVal-2 | 0.35 ± 39% | 0.57 ± 38% | 0.66 ± 36% |
| CCMI-1 | 0.32 ± 27% | 0.53 ± 29% | 0.58 ± 32% |
| CCMVal-2 and CCMI-1 | 0.35 ± 37% | 0.54 ± 32% | 0.64 ± 31% |





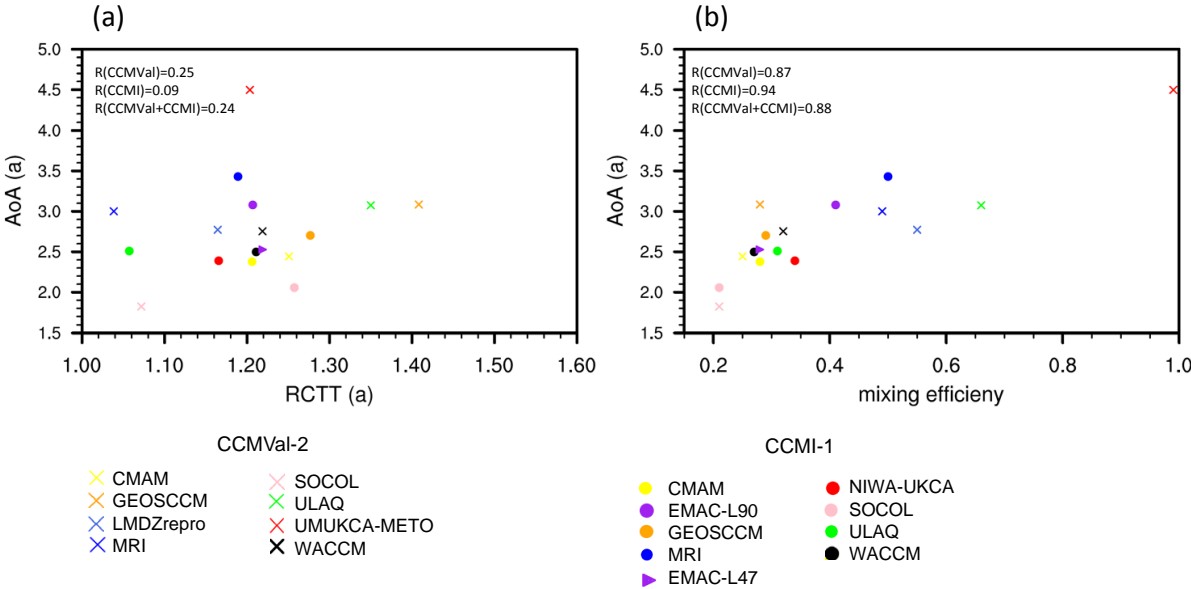

**Figure 6.** Scatter-plot showing the relationship between mean tropical (20°N-20°S) AoA and (a) mean tropical RCTT, and (b) mixing efficiency. CCMVal-2 models are represented by cross symbols and CCMI-1 models by filled dots, except EMAC-L47, which is represented by a triangle. Values are all given at 10 km above the tropical tropopause. The corresponding correlation coefficients R are given within the individual panels.

## 5 Discussion

In the last section, we showed that differences in the simulation of AoA in different models are strongly determined by differences in the mixing efficiency, i.e. the relative enhancement of AoA by any mixing processes in the model. In the formulation of the TLP model, the mixing efficiency is defined as the relative strength of horizontal mixing between the up- and downwelling regions. An independent measure of the relative role of horizontal mixing and mean transport is the ratio of mean potential vorticity (PV) to the meridional PV gradient ($dPV\,dy$). Details for the calculation of the PV gradients diagnostic are given in Garny et al. (2014). The spread of mixing efficiencies across the models is, however, only weakly correlated to the PV gradient diagnostic with highest correlations found in mid-latitudes at 450 K and correlation coefficients of about 0.5 (not shown). This weak correlation indicates that other processes than horizontal mixing play an important role in determining the



mixing efficiency. In the following we present a detailed discussion of the possible effects of different processes on the mixing efficiency.

In Section 5.1 we will focus on the impact of vertical dispersion and in Section 5.2 on the impact of model-dependent representations of numerics (e.g. advection scheme and resolution) and dynamics (unresolved wave forcing).

## 5.1 Impact of vertical dispersion on AoA profiles

According to the TLP model formulation, the age difference between tropics and mid-latitudes ($\Delta AoA$) is a function of the tropical vertical velocity ($w^*$), but independent of horizontal mixing (Neu and Plumb, 1999): $\Delta AoA = (1 + \frac{1}{\alpha})H\frac{1}{w^*}$. Here, H is the constant scale height and $\alpha$ is the ratio of the mass of air in the tropics to the mass of air in the mid-latitudes. This solution is only valid when vertical diffusion is neglected. As this is not necessarily a good assumption, the vertical velocity calculated from the AoA difference will be a tracer dependent "effective vertical velocity" in the tropics ($w_{eff}$). "Effective" refers to the effective vertical transport of the regarded tracer (i.e. AoA) that is consistent with the TLP model. The effective vertical velocities calculated from age differences (AoA difference see Fig. 7d) from one model (EMAC-L90) are compared to the actual tropical mean residual vertical velocity ($w^*$) in Fig. 7a. In particular in the lower stratosphere, the effective vertical velocity (black line) calculated from the age difference overestimates the actual vertical velocity $w^*$ (black dashed line). Note that in all the models analyzed in this study, the effective vertical velocity is similar to or larger than $w^*$ (not shown), as was also shown for the CCMVal-2 models in SPARC CCMVal (2010) (their Fig. 5.6).

Vertical diffusion (or more generally, any process causing vertical disperison) reduces the AoA difference. As discussed in Neu and Plumb (1999) and in Linz et al. (2016) (for isentropic coordinates). In the following, the TLP model is modified by including vertical diffusion (calculated as Lagrangian random walk model, see Sec. 3.2). Fig. 7b and c shows tropical and mid-latitude AoA profiles simulated with the TLP model given the vertical velocity profiles from one CCM (EMAC-L90). Profiles are given in height coordinates above the tropical mean tropopause. The Lagrangian TLP model without diffusion ($K = 0$, gray line) reproduces the analytical solution of the TLP model (mixing efficiency as estimated with the method described above). The tropical AoA profile from the TLP model is close to the AoA profile from the full CCM (black line), but mid-latitude AoA of the CCM is overestimated by the TLP model without diffusion between 0-8 km above the tropical tropopause. As the vertical gradient of AoA is positive everywhere, vertical diffusion acts to reduce the vertical gradient and thus reduces AoA. Introducing vertical diffusion in the extratropics ($K_{ML} = 0.2 m^2 s^{-1}$) in the TLP model reduces extratropical AoA in the region of 0-8 km above the tropopause (red line), and weakly influences tropical AoA. Tropical vertical diffusion (with vertical diffusivity $K_{Tr} = 0.2 m^2 s^{-1}$) leads to younger air in the tropics, and this signal is propagated into the mid-latitudes (green line). Adding vertical diffusion in both regions ($K = 0.2 m^2 s^{-1}$) combines the effects of tropical and extratropical vertical diffusion (not shown). The effective vertical velocities derived from the TLP model with extratropical diffusion roughly match the effective vertical velocities from the CCM (see black and red line in Fig. 7a). This simple experiment with the TLP model thus indicates that the deviations of the effective vertical velocities (derived from age gradients) from $w^*$ can be explained by vertical dispersion, which in particular leads to a reduced vertical age gradient in the extratropical lower stratosphere. The differences between the $w_{eff}$ and $w^*$ varies across models (not shown), i.e., in some models AoA is more strongly modified





by vertical dispersion than in others. In the simplified and conceptual TLP model, a constant vertical diffusivity was prescribed to illustrate the effects of any processes that act like vertical diffusion have on the AoA profile. In the full CCMs, a number of processes might contribute to the vertical dispersion. In most models, the vertical resolution is high enough to resolve some gravity waves (or Mixed-Rossby-Gravity waves), that lead to resolved vertical dispersion. Furthermore, as we use (log-)p co-

ordinates also adiabatic mixing is in parts projected to the vertical axis. Nevertheless, diffusion due to unresolved processes and numerical diffusion (see also next section) contribute to varying degrees to vertical dispersion. Linz et al. (2016) estimate a lower stratospheric diffusivity of $K = 0.1 m^2 s^{-1}$ based on isentropic coordinate diagnostics. This value is consistent with earlier estimates (e.g., Sparling et al. (1997)). However, it is important to note that some vertical mixing is quasi-adiabatic and therefore implicitly included in isentropic (= adiabatic) coordinates. Glanville and Birner (2017) find a much enhanced

contribution to lower stratospheric water vapor transport due to vertical diffusion in pressure coordinates.

From the discussion of the effects of vertical dispersion on AoA the following conclusions can be drawn: 1) The AoA difference in the lower stratosphere is not a good measure of tropical vertical residual circulation velocities, or in other words, vertical dispersion cannot be neglected. At higher altitudes (above about 10 km above the tropical tropopause, i.e. about 26 km or 30 hPa) the age difference is a better measure of tropical residual circulation velocities. This result is in agreement with

Linz et al. (2016). 2) The mixing efficiencies derived for the models will bear non-negligible information of vertical dispersion, and are not necessarily good measures of the relative strength of horizontal mixing. As the strength of vertical dispersion differs from model to model and thus has varying influence on the mixing efficiency, the spread in the mixing efficiencies across models cannot be related to differences in horizontal mixing alone (i.e. the correlation to the PV gradient diagnostic is

weak, see above). When calculating mixing efficiencies based on the effective vertical velocities (that include the effects of vertical dispersion), the spread in those modified mixing efficiencies relates better to horizontal mixing as measured by the PV diagnostic (with a correlation coefficient of about 0.77 at 450 K in mid-latitudes), as the effective vertical velocities implicitly include the effects of vertical dispersion. In other words, the mixing efficiency diagnosed from $w^*$ is a measure of the overall effects of both horizontal and vertical mixing.

## 25  5.2  Model characteristics that can influence mixing

In this section, we discuss dynamical and numerical model characteristics which have the potential to influence horizontal, vertical and numerical mixing. First, we analyze the possible role of the models' dynamics on horizontal mixing. As mentioned above, the dissipation of wave energy in the stratosphere largely controls the BDC. This wave energy comes from resolved planetary and synoptic waves as well as from unresolved gravity waves. Butchart et al. (2011) found an approximate ratio

of 70% EPFD and 30% GWD (20% NOGWD and 10% OGWD) that drives tropical upwelling at 70 hPa in the CCMVal-2 models. However, this ratio differs largely between various models. Cohen et al. (2013) suggest that due to compensation effects between the different wave types, the impact of the differences in gravity wave perturbation on the total circulation is reduced. Hence, models tend to agree more on the total strength of the circulation than on individual components. Mixing, however, is influenced differently through the two wave types. Rossby-wave breaking predominantly causes mixing and stirring in the

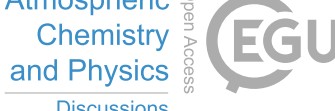

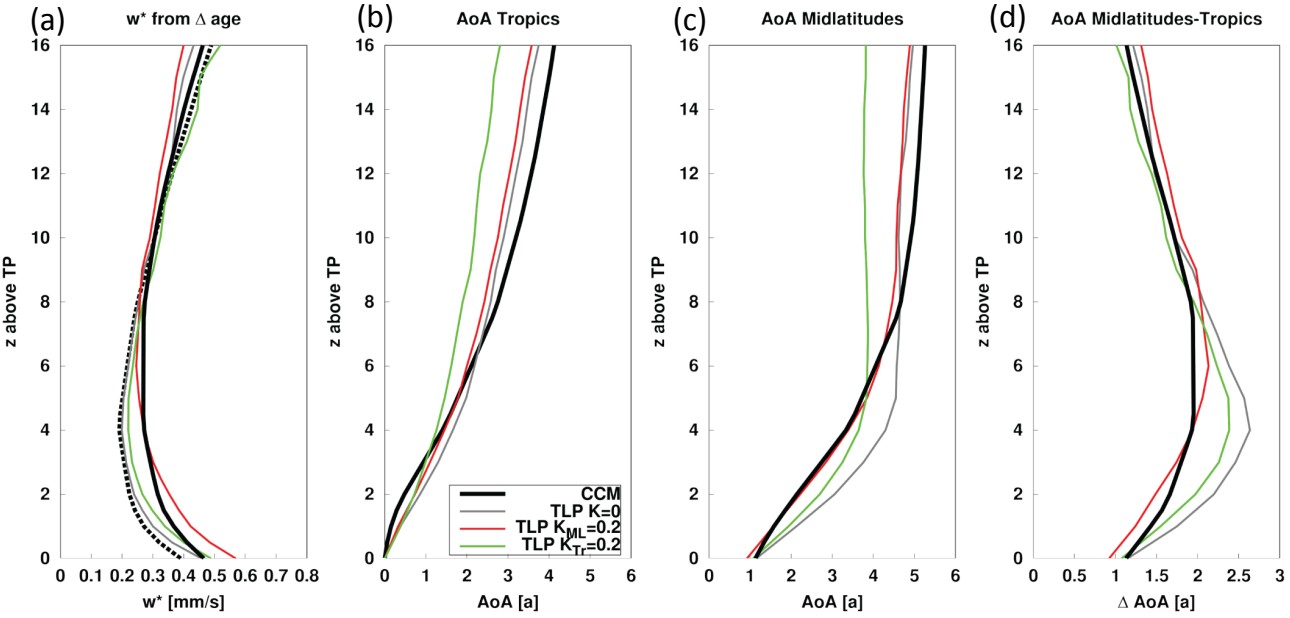

**Figure 7.** (a) Tropical mean (20°N-20°S) vertical residual velocities (black dashed) from one model (EMAC-L90) and effective tropical velocities derived from the tropics-to-midlatitude age difference in EMAC-L90 (black solid), in a TLP model driven by vertical velocities from EMAC-L90 and without diffusion (gray), with vertical diffusion of K=$0.2 m^2 s^{-1}$ in the tropics (green) and the extratropics (red). (b) Tropical (20°N-20°S) AoA profiles from EMAC-L90 (black solid) together with AoA profiles simulated by a TLP model with no vertical diffusion (gray line, identical to analytical TLP solution used to derive the mixing efficiencies), with vertical diffusion of K=$0.2 m^2 s^{-1}$ in the tropics (green) and the extratropics (red). (c) For mid-latitude AoA profiles (35°-45°N and 35°-45°S). (d) Difference between mid-latitude and tropical AoA profiles.

horizontal, while dissipation of gravity waves mainly leads to vertical mixing. Furthermore, gravity waves are parameterized in the models, and effects of mixing on tracers are usually not explicitly included in the parameterizations. Thus, while all wave types drive residual transport, GWs do not cause horizontal mixing in the same way as resolved waves do. A resulting hypothesis is that the ratio of Rossby-wave forcing to overall wave forcing influences the strength of horizontal mixing and





thus the mixing efficiency. This means that the models' ratios between EPFD and total wave forcing (EPFD+GWD) could be related to their mixing efficiencies, which could at least partly explain the AoA differences between the models.

Fig. 8 shows the climatological ratio of resolved wave drag divided to the total wave drag between 100 and 10 hPa for the CCMVal-2 REF-B1 and for the CCMI-1 REF-C1 simulations. Note that compared to the previous sections, fewer models are

5    included in this analysis because the EPFD and GWD data are not provided for all models. In the lower stratosphere, all models indicate strongest GWD contribution (low ratios), thus, here gravity wave forcing contributes strongest to the overall forcing of the residual circulation in the analyzed height range. Towards higher altitudes (10 hPa), the impact of gravity waves decreases, before it increases again strongly above 1 hPa (not shown). Three models are presented twice in the figure, once the CCMI-1 and once the CCMVal-2 simulation. The EMAC model is also presented twice, but once for the simulation with 90 layers in

10    the vertical and once with 47 layers. The two CMAM simulations show very similar wave type ratios, the two GEOSCCM simulations have a similar vertical structure, but with an offset. The MRI simulations differ vastly. Note that in none of these models, any of the gravity wave parametrization schemes have been changed from one model inter-comparison project to the other. The vertically higher resolved EMAC model has a more compact region of low wave type ratio in the lower stratosphere but otherwise the two simulations show similar results.

15    In general, the wave type ratios of the different models show a considerable spread. At 70 hPa for example, it ranges from around 0.55 in the SOCOL (CCMVal-2) simulation to around 0.9 in the GEOSCCM (CCMVal-2) simulation.

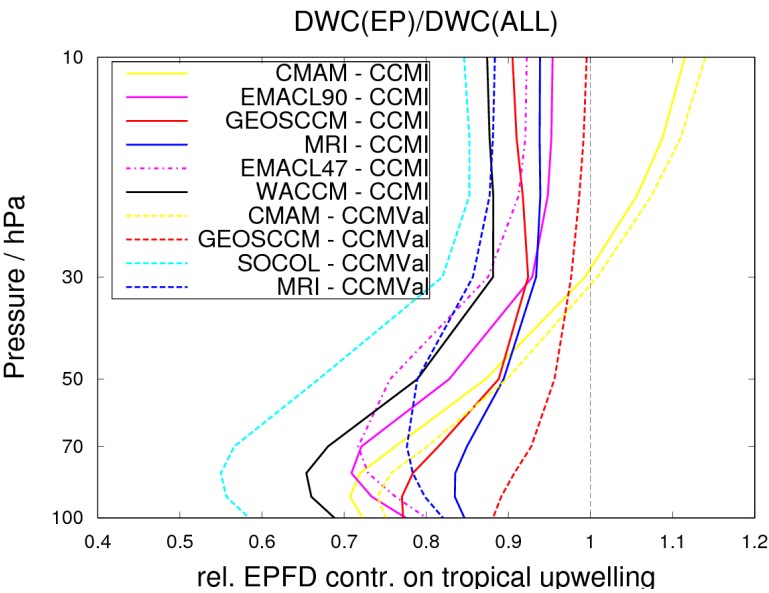

**Figure 8.** Relative EPFD contribution on tropical upwelling between 30°N-30°S as function of pressure for all CCMVal-2 and CCMI-1 models providing the data for this analysis.

As explained above, a larger ratio of resolved to parameterized wave forcing in the region where wave breaking leads to transport across the subtropical barrier causes stronger horizontal mixing and therefore, leads to additional aging by mixing





of stratospheric air. However, we found no clear correlation (ranging from 0.16 to 0.56 depending on altitude) between the wave type ratio and the mixing efficiency throughout this altitude range. The hypothesis that differences in mixing efficiency can be explained by differences in wave driving therefore has to be rejected. For two of the three models that appear twice in the statistics (CMAM and GEOSCCM), the mixing efficiency increases while the EPFD wave type ratio decreases from one

model version to another. This behavior also stands in contrast to our possible physical explanation. Rossby-waves can have a strengthening or weakening effect on the subtropical transport barrier depending on latitude and height of their location of dissipation. This may be the reason why the wave type ratio is apparently not a good measure for the mixing between tropics and extratropics. However, the sample size of the available data is too small to statistically draw robust conclusions, so more data could possibly still impact the results. As for now, however, this attempt does not provide the potential to help explaining

the AoA differences between the models.

As discussed in detail in section 5.1 vertical mixing and diffusion (both resolved and unresolved) influences AoA (and thus the mixing efficiency). Furthermore, numerical diffusion can also enhance horizontal mixing. Dietmüller et al. (2017) presented a method to separate resolved and unresolved mixing (including both vertical and horizontal unresolved mixing), by explicitly

calculating the contribution of subgrid-scale mixing to aging by mixing (termed 'aging by diffusion'). Their study showed that aging by diffusion is positive in most regions, indicating that horizontal diffusion dominates (as vertical diffusion would lead to a reduction in AoA). The calculation as performed in Dietmüller et al. (2017) requires the full 4-dimensional fields of dynamical quantities and AoA, which were not available for the CCMVal-2 and CCMI-1 models. Therefore, we can only discuss the possible differences in subgrid-scale mixing between the models qualitatively. The two factors that most likely contribute

to subgrid-scale mixing are the advection scheme and the horizontal and vertical resolution.
First we discuss the possible role of the model's advection schemes (see Table 1). The study of Eluszkiewicz et al. (2000) showed that AoA is very sensitive to the advection scheme used to integrate the tracer continuity equation. Semi-Lagrangian schemes are known to be overly diffusive, whereas the most accurate advection schemes are the finite volume and flux-form schemes. However, the more recent study of Eyring et al. (2006) showed that there are only small differences in AoA between

spectral and flux-form advection schemes, thus errors associated with spectral advection do not accumulate (Shepherd, 2007). If we link the mixing efficiency obtained for the CCMVal-2 and CCMI-1 models to their advection scheme, we find that mixing efficiency for the more accurate advection schemes (FFSL, FFEE) range from 0.21 to 0.41 (0.66 ULAQ) and for the more diffusive advection schemes (SP and SL) from 0.21 to 0.5 (0.9 UMUKCA-METO). In the SOCOL model the advection scheme was changed from SL in CCMVal-2 to FFSL in CCMI-1. However, the mixing efficiency does not change between the two

model versions. Thus, based on this information, we cannot find a systematic relationship between mixing efficiency and the different advection schemes, however as noted before the simple size is very small. Moreover, models can use the same advection scheme, but additional explicit diffusion is added, or in SL schemes higher order interpolation is possible, thus the model's advective transport can differ although using the same type of advection scheme. For example, the UMUKCA-METO model and its predecessor model NIWA-UKCA both use the same SL advection scheme, but with different polynomial interpolation.

The CCMI model NIWA-UKCA used optimized settings governing transport and advection by a higher order of interpolation.





This likely strongly reduces horizontal numerical diffusion, and thus leads to lower AoA (and a smaller mixing efficiency) in NIWA-UKCA.

Second, we address whether the increase in spatial resolution, which is apparent for many CCMs since CCMVal-2 (see the models' horizontal and vertical resolution in Table 1), has an impact on the mixing efficiency. Rind et al. (2007) showed that horizontal resolution (truncation error) has little impact on AoA, whereas a fine vertical resolution leads to higher AoA throughout the stratosphere. Faster inter-hemispheric transport and slower mixing into and out of the stratosphere cause this behavior. The models CMAM, MRI, SOCOL and ULAQ have increased their horizontal resolution since CCMVal-2, and the models MRI and ULAQ also their vertical resolution. Moreover for the model EMAC in CCMI-1 two different vertical resolutions are available. The ULAQ model is the only model that substantially changed vertical and horizontal resolution (see Table 1). The coarse resolution (in particular very low horizontal resolution) in the ULAQ REF-B1 simulation indicates that the transport barriers at the edge of the tropics and at the polar vortex are likely not reproduced very well (see also SPARC CCMVal, 2010). The quite large mixing efficiency of 0.66 in CCMVal-2 significantly improved with the higher resolution in CCMI-1 (to 0.31). The fact that ULAQ AoA in CCMVal-2 was in a similar range as the other models, might well be due to compensation effects of vertical and horizontal numerical diffusion on AoA. This hypothesis is also supported by the PV gradient of ULAQ CCMVal-2 simulation, which lies far outside of the model range (figure not shown here). Regarding the two EMAC simulations within CCMI-1, the version with higher vertical resolution has a higher AoA (see Fig. 2). The finding that higher vertical resolution leads to older air, can also be seen in the SOCOL model sensitivity simulations with different vertical resolutions (Revell et al. (2015) and A. Stenke, personal communication, 2017). The mixing efficiency in the EMAC simulation with lower resolution is reduced compared to the higher resolved model (mixing efficiency 0.41 for EMAC-L90 vrs. 0.29 for EMAC-L47). However, this is likely an effect of vertical numerical diffusion rather than changes in the horizontal mixing strength.

In general the results presented here suggest that the vertical resolution affects AoA and mixing efficiency, as seen in the EMAC and SOCOL sensitivity simulations and also for the ULAQ model. However, except ULAQ, the only model that changed vertical resolution from CCMVal-2 to CCMI is MRI, all other models only have changes in the horizontal resolution, which at this high resolution models used might not play a big role (in agreement with Rind et al. (2007)). For all other models with smaller changes in resolution than in ULAQ, no clear effect on the mixing efficiency could be detected.

The various factors that likely influence a models subgrid mixing or diffusion are hard to disentangle for the given set of models. Additional sensitivity studies with one given model would be necessary to analyze the role of the different factors (i.e. advection scheme, horizontal and vertical resolution).

## 6  Summary and Conclusions

This study analyzes climatological AoA of various stratosphere-resolving CCMs, which participated in the model inter-comparison projects CCMVal-2 and CCMI-1, in order to investigate the causes of the differences in AoA among the models.



We showed that the tropical and mid-latitude AoA profiles of most examined models have younger air compared to observations, but most AoA profiles lie within the uncertainty of values derived from observations. Moreover, there is a large spread in the simulated AoA between models. This result is in agreement with earlier model comparison studies (e.g. Eyring et al., 2006; SPARC CCMVal, 2010). We could not detect an improvement in the simulation of AoA from CCMI-1 models compared
to CCMVal-2. The CCMI-1 models tend to simulate younger air compared to their predecessor models. However, an exact one-by-one comparison is not possible because the forcings used in the CCMVal-2 and CCMI-1 hindcast simulations are not identical.

To better understand the AoA model differences, we investigated the processes that affect stratospheric transport and thus
AoA. Both, transport by the residual circulation and aging by mixing, influence the zonal structure and magnitude of AoA. Models agree on the zonal pattern of residual transport and aging by mixing, with mixing leading to additional aging in most of the stratosphere in all model simulations. Also the high inter-model correlation between tropical upwelling and RCTTs and the low and insignificant correlation between tropical upwelling and AoA indicates that mixing plays an important role in the simulation of AoA. The strength of tropical-to-mid-latitude mixing relative to residual transport is measured by the
mixing efficiency, a quantity that can be calculated from model data given the tropical mean AoA profile and tropical vertical residual velocities. The mixing efficiency is a measure of the relative aging by mixing in a model, independent of the strength of the residual circulation and it varies strongly between models. However the mixing efficiency measures the overall effects of mixing, as it accounts for both horizontal and vertical mixing and both resolved and unresolved mixing. We showed with the help of the Lagrangian TLP model that vertical diffusion has a significant impact on the mixing efficiency and thereby on
the structure of AoA. The consequence of this is that the mixing efficiency is not necessarily a good measure of the relative strength of horizontal mixing alone.

We could show that the model spread in the simulation of AoA is mostly caused by large differences in the mixing efficiency, because the inter-model correlation coefficient of mixing efficiency with AoA is high (0.88). Also, the correlation of
residual transport (RCTT) to AoA is low (inter-model correlation is 0.24). Thus differences in the simulated residual circulation matter less to the simulated AoA compared to the relative mixing strength. We can conclude that analyzing the model's mixing efficiency is very useful for the understanding of their differences in AoA. The values of the mixing efficiency vary strongly, ranging between 0.21 to 0.99. The multi-model mean of the mixing efficiency of the CCMVal-2 REF-B1 simulations ($\epsilon$=0.35) is similar to the one for the CCMI-1 REF-C1 simulations ($\epsilon$=0.32), but the model spread in mixing efficiency is higher in the
CCMVal-2 models (standard deviation of 39% compared to 27% in CCMI-1, without outliers).
In the SPARC CCMVal report the model performance on stratospheric transport diagnostics was qualitatively evaluated. CCMVal-2 models were graded (with grades indicating the agreement with observations) based on their mean AoA and on measures of tropical ascent and tropical-extratropical mixing derived from tracer diagnostics (see table 5.1 in SPARC CCMVal (2010)). The models with high grades in global mean AoA according to SPARC CCMVal (2010) generally also were graded
high in tropical ascent and mixing (see Fig. 5.19 in SPARC CCMVal (2010)). It was also found that the grade of tropical ascent



and mixing correlate quite strongly (see Fig. 5.20 in SPARC CCMVal (2010)). This finding is somewhat opposed to our results, where a perfect relation between residual transport and mixing would lead to the same mixing efficiency for all CCMVal-2 and CCMI-1 models. However, first the measures of tropical ascent and mixing in SPARC CCMVal (2010) is based on tracers that do not perfectly separate the processes of mixing and residual circulation and second we also do except a good relationship

between residual transport and the absolute amount of mixing (as both determined by wave driving), but the deviation from this relationship does cause the differences in the relative mixing strength (i.e. the mixing efficiency). In general, models that were graded high in SPARC CCMVal (2010) (namely CMAM, GEOSCCM, MRI and WACCM of CCMVal-2) were also found to have mixing efficiencies in the typical range (between 0.25 and 0.49) here. The ULAQ model (CCMVal-2) was graded high despite deficits found in the tropical ascent profile and rapid mixing across the subtropics. In our analysis we find a high mixing

efficiency of 0.66 for the ULAQ model, i.e. mixing relative to residual transport is strong. Thus, we can confirm that the reasonable AoA simulated in ULAQ (CCMVal-2) is likely due to compensation of deficits in tropical ascent by strong mixing. The models that obtained low grades in SPARC CCMVal (2010) and that were analyzed here are SOCOL, with very young air, and UMUKCA-METO, with very old air. For SOCOL, we found that despite slow tropical ascent, a low mixing efficiency (0.21, lowest of all models) leads to young air. For the "outlier" model UMUKCA-METO, in SPARC CCMVal (2010) slow tropical

ascent and too weak mixing was found. While weak mixing would lead to lower AoA, we show that relative to the residual circulation mixing is strong. Thus, we find that on top of a slow circulation, an excessive mixing efficiency ($\epsilon$=0.99) leads to the very old air in UMUKCA-METO. The comparison to the stratospheric transport diagnostics used in SPARC CCMVal (2010) shows that using the diagnostic of the mixing efficiency provides additional information on the ability of a model to simulate stratospheric transport. We found that the relative strength of mixing in a model can mainly explain deficits in the simulation of

AoA. However, a problem with the mixing efficiency diagnostic is the lack of observational constraints. It would be possible to define a mixing efficiency from the observational AoA profile and the vertical residual velocities estimated from the AoA gradients. However, those vertical velocities are substantially influenced by vertical diffusion and thus this mixing efficiency does not measure the same thing as the model derived mixing efficiency. Thus, we cannot identify whether deficits in the absolute circulation and mixing strength or a too strong or weak mixing efficiency are the cause for deviations in AoA from observa-

tions. Another problem might be that any errors in the calculation of AoA or RCTTs would be reflected in the mixing efficiency.

Within this study we also discussed the different dynamical and numerical model characteristics, which impact horizontal, vertical and numerical mixing. Besides vertical diffusion (section 5.1), subgrid-scale mixing likely influences the mixing efficiency. This assumption motivates a closer look at the possible impact of the models' different advection schemes as well as

horizontal and vertical resolution on sub-grid-scale mixing (section 5.2). The results suggest that the vertical resolution affects AoA and mixing efficiency, as seen from EMAC and SOCOL sensitivity simulations with different vertical resolution (for EMAC the mixing efficiency increases from 0.29 to 0.41 with higher resolution, for SOCOL the sensitivity simulation was not available within CCMI-1). Moreover, for the ULAQ model a substantial increase in the resolution (both horizontal and vertical) between CCMVal-2 and CCMI-1 strongly reduced the mixing efficiency (from 0.66 to 0.31). We did not find a systematic re-

lationship between mixing efficiency and the models different advection schemes. In general no systematic attribution of AoA





differences to advection schemes or resolution could be made. This is because more than one parameter has been changed between the simulations. Furthermore, we presented that the relative contribution of resolved versus parametrized wave forcing of the circulation is very different among the models. Since resolved Rossby-wave forcing induces strong horizontal mixing, parametrized GW forcing induces no mixing and both drive the residual circulation, this might have an influence on the mixing

efficiency. However, since the correlation of modeled wave type ratio with the mixing efficiency is very low, this attempt does not provide the potential to help explaining the AoA differences between the models. Concluding we can say that we found some evidence for the differences in mixing efficiency. However, overall, dedicated sensitivity studies with at least one given model system will be necessary to better determine the role of possible causes for the spread in the mixing efficiency (e.g. differences in resolution, advection scheme, GW drag).

Previous studies showed that within one model, the mixing efficiency remains constant also in a changing climate (Garny et al., 2014). If this is true for all models, any changes in the residual circulation will be related linearly to changes in AoA (as also suggested by Austin and Li, 2006). The different values of the mixing efficiency in models would then modulate the relative increase in AoA by increase in the residual circulation. In a follow-up study, we will focus on AoA trends in the CCMVal-2 and CCMI-1 future change scenario simulations and investigate how the mixing efficiency in the analyzed models evolves in a

changing climate, and possible processes for changes in the mixing efficiency will be discussed.

*Author contributions.* Simone Dietmüller, Roland Eichinger and Hella Garny made substantial contributions to conception and design, analysis and interpretation of the data. Moreover they participated in drafting the article. Thomas Birner and Harad Bönisch contributed to discussion on the content and the structure of the paper. The other authors contributed information pertaining to their individual models and helped revise this paper.

Data: All data of CCMVal-2 and CCMI-1 used in this study can be obtained through the British Atmospheric Data Centre (BADC) archive (ftp://ftp.ceda.ac.uk). CESM1-WACCM data have been downloaded from http://www.earthsystemgrid.org. For instructions for access to both archives see http://blogs.reading.ac.uk/ccmi/badc-data-access. MIPAS Data are available from https://www.imk-asf.kit.edu/english/308.php (after registration).

The authors declare that they have no conflict of interest.

*Acknowledgements.* This study was funded by the Helmholtz Association under grant VH-NG-1014 (Helmholtz-Hochschul- Nachwuchs-forschergruppe MACClim). We acknowledge the modeling groups for making their simulations available for this analysis, the WCRP SPARC Chemistry-Climate Model Validation (CCMVal) Activity as well as the SPARC/IGAC Chemistry-Climate Model Initiative (CCMI) project for organizing and coordinating the model data analysis activity, and the British Atmospheric Data Center (BADC) for collecting and archiv-

ing the CCMVal and CCMI model output. Moreover we want to acknowledge that the EMAC simulations were done within the project ESCiMo (Earth System Chemistry integrated Modelling), a national (German) contribution to the Chemistry Climate Model Initiative, and have been performed at the German Climate Computing Centre DKRZ through support from the Bundesministerium für Bildung und Forschung (BMBF). Moreover we thank Florian Haenel and Gabriele Stiller for providing us the MIPAS AoA data and also Andreas Engel for providing us the in-situ tropical AoA data.





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
