# Peer review of "Quantifying the effect of mixing on the mean Age of Air in CCMVal-2 and CCMI-1 models"

_Atmospheric Chemistry and Physics, 2017_

## Referee Comment (RC1) · Anonymous Referee #1 · 18 Jan 2018

This study presents a thorough comparison of the stratospheric mean age of air among models participating in two comprehensive model intercomparisons (CCMVal and CCMI). They show that there is a slight degradation in the mean age in the more recent CCMI model ensemble, which is biased young relative to observations. Furthermore, the authors show that most of the spread among models is related to differences in mixing, which the authors define in a particular sense as the residual between the mean age and the (idealized) transport times associated only with advection by the residual mean circulation. While I do struggle a bit with defining an "aging by mixing" simply as a residual (as opposed to more conventional approaches using the TLP model) I think this study represents an important contribution. I commend the authors for including both CCMVal and CCMI results in the study, which highlights persistent

problems in stratospheric transport despite (or because of) the incorporation of higher model tops, higher vertical resolution, more sophisticated gravity wave drag parameterizations, etc. I also commend the authors for utilizing a Lagrangian TLP model to illustrate the impacts of vertical diffusion on the mean age. I recommend publication once the comments outlined below are addressed. There are several grammatical errors and awkward phrasing that need to be fixed (some of which are delineated below but should be caught after a more careful reading of the text by the authors).

Minor Comments:

1) It is misleading to say that the approaches in Garny et al. (2014), etc. (lines 11-12, Page 3) first demonstrated that mixing can enhance the ages in the stratosphere. A similar claim is made in line 24 on Page 7. This was first shown by Hall and Plumb (1996) (and studies thereafter), albeit in a theoretical context. This is a major oversight and should be corrected in the text.

2) Some more discussion (with caveats) is needed concerning the trajectory calculations used to infer the "residual circulation transit times." In particular, what contribution do differences in tropopause height among the models contribute to these inferred times? One possibility would be to approximate this by repeating the calculation for one model (with varying tropopause definitions) and/or commenting on previous studies that have looked into the robustness of these calculations.

Other comments:

1) line 5, page 1: "by both, mean transport" -> "by both mean transport by"

2) line 5, page 1: please replace "transport along the residual circulation" with "transport by..."

3) line 16, page 2: "it is the only possible observation-based measure of the BDC" -> This is not true. Previous studies have constrained related aspects of the transport circulation (the age spectrum) using combinations of CFCs (e.g. "Estimation of

stratospheric age spectrum from chemical tracers" by Schoeberl and Douglass (2005)). Please rephrase.

4) line 9, page 3: remove comma after "both"

5) line 3, page 7: "Stratospheric mean age of air is defined as the residence time" -> The mean age and mean residence time are distinct quantities, as outlined in Holzer, Orbe and Primeau (2012). Please replace "residence time" with "transit time".

Holzer, M., C. Orbe and F. W. Primeau. "Stratospheric mean residence time and mean age on the tropopause: Connections and implications for observational constraints." Journal of Geophysical Research: Atmospheres 117.D12 (2012).

6) line 13, page 7: Please fix the latex error.

7) line 4, page 8: "At the same time" -> I'm not sure this is the right clause to use.

8) line 5-10, page 10: It is not clear how you're defining the mixing efficiency. It would be easiest to write out the analytic expression right after the sentence beginning with "The mixing efficiency can be calculated ...". Something along the lines of the discussion in line 11 on page 17, but here.

9) line 29, page 9: Indeed the discrepancy between the ACCESS and NIWA models is concerning!

Figures 1 and 2: It would be good to align the figures as much as possible so that the reader can easily see the projection/degradation of the mean ages among the same models (e.g. move ULAQ, GEOSCCM, SOCOL rows).

---

## Referee Comment (RC2) · M. Linz (Referee) · 2 Feb 2018

Review of "Quantifying the effect of mixing on mean Age of Air in CCMVal-2 and CCMI-1 models"

In this paper, the authors present a thorough and well thought out analysis of the contributions of mixing to stratospheric age of air in both the most recent generation of coupled chemistry climate models and the previous generation. This work is an important contribution to understanding model differences in tracer transport and in their representation of the circulation. They find that the residual circulation differences are not the primary driver of the age differences between models—rather the mixing efficiency differences are critical. They make an heroic attempt to understand what leads to mixing efficiency differences in the models, concluding that there are substantial difficulties in this analysis in an intermodel comparison.

This is an excellent, valuable paper, and I enjoyed reading it.

I have some relatively minor points for the authors to address. I would also highly recommend a thorough read-through for grammar, as there are misplaced commas (etc.) throughout.
P1L7-8: AoA is not a measure of the strength of the BDC, or at least it isn't depending on your preferred definition of the BDC (some would equate BDC with residual circulation). I would recommend removing this and just say how it's defined.
P3L11: Please include some more of the historical mixing literature. E.g. Newman et al. 1986, the eddy diffusivity literature of Nakamura 1996 and related observational follow-ups by Haynes and Shuckburg (2000) and Allen and Nakamura (2001).
P5L8: "Basically … schemes" I found this confusing.
P7L3: mean age is definitely not the same as the residence time
P7L13: ?
P7L23: transport by the residual circulation, not through the residual circulation
P8: please clarify exactly how you calculate mixing efficiency
P8L30: allows linear separation of …
P9L26-30…/Fig. 3: I'll send you age from MLS N2O which has been calculated using Andrews (2001) relationship (along with the citations). This is an additional new observational constraint if you would like to include it. Let me know if you need any more information about it.
P10L20: Perhaps also include Kovacs et al. 2017 (http://eprints.whiterose.ac.uk/111819/1/acp-17-883-2017.pdf)
P16: Please clarify what was done for this analysis. If the correlation is really of the climatologies, one interpretation of that correlating the mean January upwelling to the mean January RCTT, in which case this is answering the question of whether the seasonal cycles are correlated. Generally though note that it makes sense for age to be less well correlated than RCTT, since age is independent and RCTT depends on the residual circulation, which is used to calculate the upwelling as well.
P17L10: This discussion of mixing efficiency in terms of AoA-RCTT is confusing. I thought Garny et al. 2014 showed that they are related but not mathematically equivalent, because of the dependency of aging by mixing on the vertical velocity.
Generally when writing about mixing efficiency, it is assumed that models far from the multimodel mean are "worse". Since we do not know that the multimodel mean is right, I would be careful with this language.
P18L10: awk. Perhaps "mixing leads to different magnitudes in the relative enhancement of AoA" or something like that.
P18L20: "exemplary" not sure what you mean here

5.1: How was \Delta(AoA) calculated? what were the tropical edges? This matters in isentropic coordinates, and I'd expect it to matter even more for pressure coordinates. If this is with 20 degree tropics, I would ask that the calculation be redone for 35 degree or turnaround latitude tropics, to see if that makes a difference.

P21L25: I think you mean \Delta(AoA)

P22L13-14: So this is not quite true. AoA difference is a biased measure in the lower stratosphere (and as you note above, it's better in isentropic coordinates than in pressure coordinates), but that doesn't mean it's useless even in the light of vertical diffusion being important.

P23L3-4: "A resulting hypothesis…" It's a shame to bury this sentence in the middle of the paragraph. Start a new paragraph, perhaps?

P24L11-13: Awkward phrasing

P25L31: "different advection schemes with this small sample size." would be more clear

P26L12: "improved" should be "is significantly closer to the multimodel mean"

P27L21: Perhaps add a comment here about diagnosing epsilon differently at different levels? e.g. Ray et al. 2010? Not necessary, but potentially interesting, since the neglect of vertical diffusion isn't as problematic higher up.

P27L23: "showed" rather than "could show"

P27L26: "models'" rather than "model's"

P28L5: "as both are driven by wave driving" or some such

P28L16: "a large" rather than "an excessive" Again, we should reserve judgment until we can actually compare

P29L2: "Demonstrated" rather than "presented"

P29L5-7: "… modeled wave type ratio with the mixing efficiency is very low, the difference in models resolved and parameterized waves does not explain the AoA differences…"

Figures: Stylistically, I think it would help a lot if font sizes were consistent and larger

Figure 6 (b) missing a "c" in mixing efficiency. The R values are hard to read.

Figure 7: choose a different color besides green (or make it dot dash or something) to be colorblind friendly

---

## Referee Comment (RC3) · E. Ray (Referee) · 2 Feb 2018

This paper examines the age of air and mixing, both horizontal and vertical, among a suite of chemistry-climate model output spanning two projects over the past decade. This is a very nice analysis that highlights key aspects of mixing that lead to discrepancies among modeled age of air. It's surprising to see the importance of vertical mixing and diffusion as compared to horizontal mixing on the mean ages. Ideally this work will point to areas of focus to help reduce the persistent spread in the modeled age of air.

The results are well organized and described and the topic is appropriate for ACP. I recommend publication with consideration of the specific comments below.

Specific comments:

[Figure]

Pg. 2, line 11: switch "also" and "be"

Pg. 7, line 13: "Karlsruhe, Germany" seems to be a mistake

Pg. 8, line 30: remove "allows" and change to "separates"

Pg. 10 and references: Problem with the "Ray and Andrews, 2017" reference.

Pg. 11, line 22: add "of" after "question"

Pg. 16: I was unsure in reading the first paragraph of Section 4.2 whether the correlations shown in Figure 4 were done for all of the models or not. At the end of the paragraph you do mention the sample size of 17 models but I would mention that up front to make it clearer. How much variability in the correlations is there between models?

Pg. 16, lines 22,23: add commas after "RCTTs" and "tropical pipe"

Pg. 17, line 16: add "a" before "measure"

Pg. 18, line 10: change to "different" and remove "strong relative"

Pg. 18, line 20: remove "exemplary"

Pg. 22, line 5: remove "also" and add comma in that space, change "in parts" to something like "partly"

Pg. 24, line 3: change "to" to "by"

Pg. 24, line 6: remove "here"

Pg. 24, lines 8-9: ". . .twice in the figure, once each for the CCMI-1 and CCMVal-2 simulations."

Pg. 24, figure 8: A more descriptive title on the figure would be helpful.

Pg. 25, lines 2-3: The statement that wave driving differences can't explain mixing efficiency differences might be too strong based on the small sample size statement

made later in the paragraph.

Pg. 25, line 9: change to "explain"

Pg. 25, line 12: change to "influence"

Pg. 25, line 31: change to "sample"

Pg. 25-26: The discussion of the model's advection schemes and resolution in these two pages could be shortened in my opinion. The results are interesting but the discussion section is long. Since there is no systematic relationship found among the advection schemes it would be sufficient to just mention that without going into the details.

Pg. 27, line 23: remove "could" and change to "showed"

Pg. 27, line 25: add comma after "Thus"

Pg. 28, line 3: change "is" to "were"

Pg. 28, line 4: change "do" to "did", change "also do except" to "expect"

Pg. 28, line 6: remove "does" and change to "caused"

Pg. 29, line 6: change to "explain"

---

## Author Comment (AC1) · 12 Apr 2018

We thank referee #1 for the positive and constructive comments on our manuscript. Below, the manuscript is changed taking into account his questions  and comments. The changed manuscript, with changes high-lighted, is attached to the reply.

Note that during the review process we found, that the calculation of w* is treated inconsistent within the different models, as in some models a fixed scale height was used to transform w* from Pa/s to m/s, while in other models the actual density was used for this transformation. The different calculation methods of w* can lead to large differences in w*. To facilitate a quantitative model comparison we recalculated w* from the given v* fields. Thus we recalculated the RCTTs, as well as the mixing efficiencies, however the conclusions of the paper do not change. We now provide a supplement (attached to this reply).

**Minor Comments:**

1. It is misleading to say that the approaches in Garny et al. (2014), etc. (lines 11-12, Page 3) first demonstrated that mixing can enhance the ages in the stratosphere. A similar claim is made in line 24 on Page 7. This was first shown by Hall and Plumb (1996) (and studies thereafter), albeit in a theoretical context. This is a major oversight and should be corrected in the text.
➜ *Yes you are right, the sentence can be misunderstood. Of course Garny et al. 2014 were not the first showing that mixing does influence AoA. I wanted to say that they were the first in determining aging by mixing from global model data. To be more precise, I added some sentences (see page 3, line 10-13), where I included some historical mixing literature (diagnostic of local eddy mixing e.g. Newman et al 1986, Haynes and Shuckburgh 2000) and also as you suggested the theoretical concept studies  with idealized models that are showing that mixing enhances AoA (e.g. Hall and Plumb 1994, Neu and Plumb 1999).*

2.  Some more discussion (with caveats) is needed concerning  the trajectory calculations used to infer the "residual circulation transit times." In particular, what contribution do differences in tropopause height among the models contribute to these inferred times? One possibility would be to approximate this by repeating the calculation for one model (with varying tropopause definitions) and/or commenting on previous studies that have looked into the robustness of these calculations.
➜ *Yes that's right the RCTTs on a certain pressure layer are dependent on the tropopause height (definition). Near the tropopause that makes a significant difference. This problem we do not take into account in the RCTT calculation. However later on when determining the mixing efficiency we calculate all values relative to 100 hPa. As you can see the spread in the*

*RCTTs is still high, thus we can conclude that the spread in RCTTS is not due to tropopause. I included some discussion on this at page 11.*

**Other comments:**
1) line 5, page 1: "by both, mean transport" -> "by both mean transport by"
→ *done*

2) line 5, page 1: please replace "transport along the residual circulation" with "transport by..."
→ *done*

3) line 16, page 2: "it is the only possible observation-based measure of the BDC" -> This is not true. Previous studies have constrained related aspects of the transport circulation (the age spectrum) using combinations of CFCs (e.g. "Estimation of stratospheric age spectrum from chemical tracers" by Schoeberl and Douglass (2005)). Please rephrase.
→ rephrased

4) line 9, page 3: remove comma after "both"
→ *done*

5) line 3, page 7: "Stratospheric mean age of air is defined as the residence time" ->The mean age and mean residence time are distinct quantities, as outlined in Holzer, Orbe and Primeau (2012). Please replace "residence time" with "transit time".
Holzer, M., C. Orbe and F. W. Primeau. "Stratospheric mean residence time and mean age on the tropopause: Connections and implications for observational constraints." Journal of Geophysical Research: Atmospheres 117.D12 (2012).
→ *done*

6) line 13, page 7: Please fix the latex error.
→ *done*

7) line 4, page 8: "At the same time" -> I'm not sure this is the right clause to use.
→ *done*

8) line 5-10, page 10: It is not clear how you're defining the mixing efficiency. It would be easiest to write out the analytic expression right after the sentence beginning with "The mixing efficiency can be calculated ...". Something along the lines of the discussion in line 11 on page 17, but here.
→ *I included the analytic expression for the mixing efficiency calculation in the text now. See page 8, equation 1.*

9) line 29, page 9: Indeed the discrepancy between the ACCESS and NIWA models is concerning!

→ *Yes, I were in contact with both modeling groups, and they could not identify a difference in the model setup (since now) and the post-processing handling is identical.*

Figures 1 and 2: It would be good to align the figures as much as possible so that the reader can easily see the projection/degradation of the mean ages among the same models (e.g. move ULAQ, GEOSCCM, SOCOL rows).

→ *done*

[revised manuscript text omitted]

| GEOSCCM | ? | 2.0°x2.5°, L72 | 0.015hPa | FFSL |
| LMDZrepro | ? | 2.5°x3.8°, L50 | 0.07hPa | FV |
| MRI | ? | T42, L68 | 0.01hPa | STFD* |
| SOCOL | ? | T30, L39 | 0.01 hPa | SL* |
| ULAQ | ? | 11.5°x22.5°, L26 | 0.04 hPa | FFEE |
| UMUKCA-METO | ? | 2.5°x3.8°, L60 | 84km | SL |
| WACCM | ? | 1.9°x2.5°, L66 | 0.00005hPa | FFSL |
| **CCMI-1** | | | | |
| ACCESS-CCM | ??? | 2.5°x3.8°, L60 | 84km | SL |
| CMAM | ? | T47, L71 | 0.0008hPa | SP |
| | ? | | | |
| CESM1-WACCM | ?? | 1.9°x2.5°, L66 | 140kCCMI$_2$0171203.texm | FFSL |
| | ? | | | |
| EMAC-L90 | ?? | T42, L90MA | 0.01hPa | FFSL |
| EMAC-L47 | ?? | T42, L47 | 0.01hPa | FFSL |
| GEOSCCM | ?? | 2°x2.5°, L72 | 0.015hPa | FFSL |
| | ?? | | | |
| MRI | ? | TL159, L80 | 0.01 hPa | SL |
| | ?? | | | |
| SOCOL | ?? | T42, L39 | 0.01 hPa | FFSL |
| NIWA-UKCA | ?? | 2.5°x3.8°, L60 | 84km | SL |
| | ? | | | |

[revised manuscript text omitted]
-1 include data requests for the residual velocities $\bar{v}^*$ and $\bar{w}^*$. The $\bar{w}^*$ fields are specified to be given in units of $m/s$. However, as most models use hybrid-pressure coordinates, the vertical velocity $\bar{\omega}^*$ in $Pa/s$ is usually available and has to be transformed to $m/s$. The exact transformation of $\omega$ to w is given by:

$$\omega \approx \frac{dp}{dt} = \frac{\partial z}{\partial t}\frac{\partial p}{\partial z} = \mathrm{w}\frac{-pg}{RT} \tag{1}$$

I.e. on a given pressure level $p$ the transformation is dependent on the local temperature $T$. Alternatively, a constant scale height assumption can be made for the transformation:

$$\omega \approx \mathrm{w}\frac{-p}{H} \tag{2}$$

In the CCMI data request a scale height of H = 6950 m was suggested. Fig. 1 compares the profile of tropical mean $\bar{\mathrm{w}}^*$
from one model (EMAC) calculated with the actual density (Equ. 1, red line) and with the scale height (Equ. 2, black line). Differences in the lower stratosphere (between 100 to 70 hP) are more than 20%. Upon close inspection of delivered CCMI model output, it turned out that some model data was processed converting vertical velocities with the actual density while others used a fixed scale height. Given the large deviations of the two methods, this inconsistency has to be taken into account for a quantitative model comparison. In the following we give an outline how those inconsistencies can be detected, and how a consistent model comparison can be achieved.

Given continuity, the residual vertical velocity is linked to the meridional velocity:

$$\frac{\partial \bar{v}^*}{\partial y} + \frac{\partial \bar{\omega}^*}{\partial p} = 0 \tag{3}$$

Therefore, it is possible to deduce the residual vertical velocity $\bar{\omega}^*$ from the residual meridional velocity by vertical integration and meridional derivation. This vertical velocity $\bar{\omega}^*$ can then again be converted to $\bar{\mathrm{w}}^*$ using the given scale height. By comparing the given $\bar{\mathrm{w}}^*$ from the model output to the $\bar{\mathrm{w}}^*$ that is calculated from $\bar{v}^*$ (referred to as $\bar{\mathrm{w}}^*_{\bar{v}^*}$ in the following) it can

[Figure]

**Figure 1.** Profile of climatological mean $\bar{w}^*$ over 20N-20S for EMAC-L90. Left: $\bar{w}^*_{CCMI}$ delivered to CCMI with actual density used in the transformation (red), $\bar{w}^*_H$ with scale height used for the transformation (black) and $\bar{w}^*_{\bar{v}*}$ derived from $\bar{v}*$ (blue). Right: relative difference between $\bar{w}^*_{CCMI}$ to $\bar{w}^*_H$ (red) and relative difference between $\bar{w}^*_{\bar{v}*}$ and $\bar{w}^*_H$ (blue).

be tested whether the same scale height was used originally in the calculation of $\bar{w}^*$. As shown in Fig. 2, tropical mean $\bar{w}^*$ at 70 hPa as delivered for the CCMI models agrees with the derived $\bar{w}^*_{\bar{v}*}$ within <10% for 4 models, while in 3 models to difference is close to or above 20%. After consulting the modeling groups it turned out that the three models with the largest difference (EMAC, NIWA-UKCA, SOCOL) indeed used the actual density to calculate $\bar{w}^*$ (i.e. Equ. 1) instead of the fixed scale height.

5 This inconsistent calculation method inhibits quantitative model comparison of the delivered $\bar{w}^*$ fields, as differences due to different methods are of similar magnitude than the model differences (as can be seen from Fig. 2).

To ensure consistent treatment of model data, we used in our study the $\bar{w}^*$ fields derived from $\bar{v}^*$. While this calculation also introduces errors due to the vertical integration and meridional derivation, it is ensured that the model data is treated in the same way for all models. The error made in the derivation of $\bar{w}^*_{\bar{v}*}$ is shown for the EMAC model in the right panel of Fig. 1:

10 the difference between the $\bar{w}^*$ calculated directly from $\bar{\omega}*$ with the scale height and the $\bar{w}^*_{\bar{v}*}$ derived from $\bar{v}^*$ (blue line) is smaller 5%, i.e. much smaller than the difference due to scale height versus density transformation.

Based on those results we strongly encourage anyone working with residual velocities from multi-model comparison projects to:

1. Check if the given residual vertical velocities $\bar{w}^*$ are consistently calculated by comparison to the derived $\bar{w}^*_{\bar{v}*}$ from $\bar{v}^*$

[Figure]

**Figure 2.** Climatological mean $\overline{w}^*$ over 20°N-20°S at 70 hPa for all CCMI REF-C1 model simulations used in this study. The left bar is the $\overline{w}^*$ delivered by the models, the right bar is $\overline{w}^*{}_{\overline{v}*}$ derived from $\overline{v}^*$. The number at the top is the percentage difference.

2. If necessary, use those derived $\overline{w}^*{}_{\overline{v}*}$ values for quantitative model comparison

---

## Author Comment (AC2) · 12 Apr 2018

**Reply to Referee M. Linz (ACP-2017-1143)**

We thank M. Linz a lot for the positive and constructive comments on our manuscript. Below we summarize our answers to her specific comments. Moreover the manuscript is changed taking into account the comments (changed manuscript with changes highlighted is attached to this reply) .

Note that during the review process we found, that the calculation of w* is treated inconsistent within the different models, as in some models a fixed scale height was used to transform w* from Pa/s to m/s, while in other models the actual density was used for this transformation. The different calculation methods of w* can lead to large differences in w*. To facilitate a quantitative model comparison we recalculated w* from the given v* fields. Thus we recalculated the RCTTs, as well as the mixing efficiencies, however the conclusions of the paper do not change. We now provide a supplement (attached to this reply).

**Specific issues:**

P1L7-8: AoA is not a measure of the strength of the BDC, or at least it isn't depending on your preferred definition of the BDC (some would equate BDC with residual circulation). I would recommend removing this and just say how it's defined.
➙ *Done.*

P3L11: Please include some more of the historical mixing literature. E.g. Newman et al. 1986, the eddy diffusivity literature of Nakamura 1996 and related observational follow-ups by Haynes and Shuckburg (2000) and Allen and Nakamura (2001).
➙ *We now include some sentences about previously used eddy mixing diagnostics (e.g. Newman et a. 1986, Nakamura 1996, Haynes and Shuckburg 2000). However these studies are looking at local mixing effects. As we are interested in the integrated effect of mixing, we focus on the "aging by mixing" diagnostic. I hope this is clear in the text now (see p.3, line 10-13). Additionally (comment of another referee) we add a sentence about the theoretical concept studies with idealized models that are showing that mixing enhances AoA (e.g. Hall and Plumb 1994, Neu and Plumb 1999).*

P5L8: "Basically … schemes" I found this confusing.

→ *rephrased the sentence to: "The GW schemes used by the different models are listed in Table S9 of …"*

P7L3: mean age is definitely not the same as the residence time
→ I *replaced "residence time" with "transit time".*

P7L13: ?
→ *Error fixed.*

P7L23: transport by the residual circulation, not through the residual circulation
→ *Corrected.*

P8: please clarify exactly how you calculate mixing efficiency
→ *I now included the analytic expression for the mixing efficiency calculation in the text (analog to Garny et al 2014). See page 8, Equation 1.*

P8L30: allows linear separation of …
→ *Done.*

P9L26-30…/Fig. 3: I'll send you age from MLS N2O which has been calculated using Andrews (2001) relationship (along with the citations). This is an additional new observational constraint if you would like to include it. Let me know if you need any more information about it.
→ *Thanks a lot for the MLS N2O data, I included them to my figures 3a, 3b and 3c. An additional sentence is included for the description of the MLS data together with the citation of Andrews 2001 and Linz et al. 2017. Moreover the data availability is included.*

P10L20: Perhaps also include Kovacs et al. 2017 (http://eprints.whiterose.ac.uk/111819/1/acp-17-883-2017.pdf)
→ *done*

P16: Please clarify what was done for this analysis. If the correlation is really of the climatologies, one interpretation of that correlating the mean January upwelling to the mean January RCTT, in which case this is answering the question of whether the seasonal cycles are correlated. Generally though note that it makes sense for age to be less well correlated than RCTT, since age is independent and RCTT depends on the residual circulation, which is used to calculate the upwelling as well.
→ *The correlation is indeed calculated from the climatologies, but also integrated over the months, i.e. the annual mean values of tropical upwelling at a level from each model are correlated to the annual mean AoA (or RCTT) value at each point. Hence, the correlation does not show seasonal cycles, but the mean. True, it is to be expected that RCTT correlated better to upwelling than AoA, the question was whether upwelling at a certain level is actually a good measure for*

*RCTT at each point (or not). We will include a few words for both these points.*

P17L10: This discussion of mixing efficiency in terms of AoA-RCTT is confusing. I thought Garny et al. 2014 showed that they are related but not mathematically equivalent, because of the dependency of aging by mixing on the vertical velocity.
→ *This is true for the difference AoA-RCTT, but here we the* **relative** *increase of AoA due to mixing (AoA-RCTT)/RCTT). If you look at equation 1 (vertical velocity does depend on height!), we can reformulate this equation (if neglecting the height dependence) to: $AoA_T = RCTT_T + eps * alpha' * (RCTT_T)$. Solving this equation 1 for epsilon will give $eps = (AoA_T - RCTT_T)/(RCTT_T)/alpha'$ (see page 8)*

Generally when writing about mixing efficiency, it is assumed that models far from the multimodel mean are "worse". Since we do not know that the multimodel mean is right, I would be careful with this language.
→ *You are right.*

P18L10: awk. Perhaps "mixing leads to different magnitudes in the relative enhancement of AoA" or something like that.
→ *rephrased*

P18L20: "exemplary" not sure what you mean here
→ *deleted*

5.1: How was \Delta(AoA) calculated? what were the tropical edges? This matters in isentropic coordinates, and I'd expect it to matter even more for pressure coordinates. If this is with 20 degree tropics, I would ask that the calculation be redone for 35 degree or turnaround latitude tropics, to see if that makes a difference.
→ *The tropical edges are 20N/S see caption of fig. 7. I now included this information to the text (same edges as in the diagnostics before). We did play around with different latitudes, coming to the same conclusion.*

P21L25: I think you mean \Delta(AoA)
→ *No, vertical gradient is actually the correct term here. As the vertical gradient is positive, vertical diffusion leads to a reduction in the gradient. However, as this sentence is too confusing (and not important), we removed it from the manuscript.*

P22L13-14: So this is not quite true. AoA difference is a biased measure in the lower stratosphere (and as you note above, it's better in isentropic coordinates than in pressure coordinates), but that doesn't mean it's useless even in the light of vertical diffusion being important.

→ *True, and we did not mean to state that the AoA difference is useless, one just has to be careful what it measures. Sentence changed to: "The AoA difference is a biased measure of the tropical vertical residual circulation velocities in the lower stratosphere, or in ..."*

P23L3-4: "A resulting hypothesis..." It's a shame to bury this sentence in the middle of the paragraph. Start a new paragraph, perhaps?
→ *I started a new paragraph now.*

P24L11-13: Awkward phrasing
→ *Sentence changed to: Note that in both model inter-comparison projects the same gravity wave parametrization schemes have been used in the respective models.*

P25L31: "different advection schemes with this small sample size." would be more clear
→ *rephrased*

P26L12: "improved" should be "is significantly closer to the multimodel mean"
→ *done*

P27L21: Perhaps add a comment here about diagnosing epsilon differently at different levels? e.g. Ray et al. 2010? Not necessary, but potentially interesting, since the neglect of vertical diffusion isn't as problematic higher up.
→ *I decided not to include a comment about the possibility of diagnosing epsilon differently at different levels (although the vertically varying mixing efficiency could lead to significant differences, as vertical diffusion, as you mention is not as problematic higher up), but I think this is too much information in this context. However I keep in mind to do further analysis to investigate this effect.*

P27L23: "showed" rather than "could show"
→ *done*

P27L26: "models'" rather than "model's"
→ *done*

P28L5: "as both are driven by wave driving" or some such
→ *done*

P28L16: "a large" rather than "an excessive" Again, we should reserve judgment until we can actually compare
→ *done*

P29L2: "Demonstrated" rather than "presented"
→ *done*

P29L5-7: "… modeled wave type ratio with the mixing efficiency is very low, the difference in models resolved and parametrized waves does not explain the AoA differences…"
→ *rephrased*

Figures: Stylistically, I think it would help a lot if font sizes were consistent and larger Figure 6 (b) missing a "c" in mixing efficiency. The R values are hard to read.
Figure 7: choose a different color besides green (or make it dot dash or something) to colorblind friendly
→ *We tried to make our figures (who are plotted with different plotting programs) more consistent and we made the font sizes larger.*

[revised manuscript text omitted]

| GEOSCCM | ? | 2.0°x2.5°, L72 | 0.015hPa | FFSL |
| LMDZrepro | ? | 2.5°x3.8°, L50 | 0.07hPa | FV |
| MRI | ? | T42, L68 | 0.01hPa | STFD* |
| SOCOL | ? | T30, L39 | 0.01 hPa | SL* |
| ULAQ | ? | 11.5°x22.5°, L26 | 0.04 hPa | FFEE |
| UMUKCA-METO | ? | 2.5°x3.8°, L60 | 84km | SL |
| WACCM | ? | 1.9°x2.5°, L66 | 0.00005hPa | FFSL |
| **CCMI-1** | | | | |
| ACCESS-CCM | ??? | 2.5°x3.8°, L60 | 84km | SL |
| CMAM | ? | T47, L71 | 0.0008hPa | SP |
| | ? | | | |
| CESM1-WACCM | ?? | 1.9°x2.5°, L66 | 140kCCMI$_2$0171203.texm | FFSL |
| | ? | | | |
| EMAC-L90 | ?? | T42, L90MA | 0.01hPa | FFSL |
| EMAC-L47 | ?? | T42, L47 | 0.01hPa | FFSL |
| GEOSCCM | ?? | 2°x2.5°, L72 | 0.015hPa | FFSL |
| | ?? | | | |
| MRI | ? | TL159, L80 | 0.01 hPa | SL |
| | ?? | | | |
| SOCOL | ?? | T42, L39 | 0.01 hPa | FFSL |
| NIWA-UKCA | ?? | 2.5°x3.8°, L60 | 84km | SL |
| | ? | | | |

[revised manuscript text omitted]
-1 include data requests for the residual velocities $\bar{v}^*$ and $\bar{w}^*$. The $\bar{w}^*$ fields are specified to be given in units of $m/s$. However, as most models use hybrid-pressure coordinates, the vertical velocity $\bar{\omega}^*$ in $Pa/s$ is usually available and has to be transformed to $m/s$. The exact transformation of $\omega$ to w is given by:

$$\omega \approx \frac{dp}{dt} = \frac{\partial z}{\partial t}\frac{\partial p}{\partial z} = \text{w}\frac{-pg}{RT} \tag{1}$$

I.e. on a given pressure level $p$ the transformation is dependent on the local temperature $T$. Alternatively, a constant scale height assumption can be made for the transformation:

$$\omega \approx \text{w}\frac{-p}{H} \tag{2}$$

In the CCMI data request a scale height of H = 6950 m was suggested. Fig. 1 compares the profile of tropical mean $\bar{w}^*$
10 from one model (EMAC) calculated with the actual density (Equ. 1, red line) and with the scale height (Equ. 2, black line). Differences in the lower stratosphere (between 100 to 70 hP) are more than 20%. Upon close inspection of delivered CCMI model output, it turned out that some model data was processed converting vertical velocities with the actual density while others used a fixed scale height. Given the large deviations of the two methods, this inconsistency has to be taken into account for a quantitative model comparison. In the following we give an outline how those inconsistencies can be detected, and how a
15 consistent model comparison can be achieved.

Given continuity, the residual vertical velocity is linked to the meridional velocity:

$$\frac{\partial \bar{v}^*}{\partial y} + \frac{\partial \bar{\omega}^*}{\partial p} = 0 \tag{3}$$

Therefore, it is possible to deduce the residual vertical velocity $\bar{\omega}^*$ from the residual meridional velocity by vertical integration and meridional derivation. This vertical velocity $\bar{\omega}^*$ can then again be converted to $\bar{w}^*$ using the given scale height. By
20 comparing the given $\bar{w}^*$ from the model output to the $\bar{w}^*$ that is calculated from $\bar{v}^*$ (referred to as $\bar{w}^*_{\bar{v}^*}$ in the following) it can

[Figure]

**Figure 1.** Profile of climatological mean $\bar{w}^*$ over 20N-20S for EMAC-L90. Left: $\bar{w}^*_{CCMI}$ delivered to CCMI with actual density used in the transformation (red), $\bar{w}^*_H$ with scale height used for the transformation (black) and $\bar{w}^*_{\bar{v}*}$ derived from $\bar{v}*$ (blue). Right: relative difference between $\bar{w}^*_{CCMI}$ to $\bar{w}^*_H$ (red) and relative difference between $\bar{w}^*_{\bar{v}*}$ and $\bar{w}^*_H$ (blue).

be tested whether the same scale height was used originally in the calculation of $\bar{w}^*$. As shown in Fig. 2, tropical mean $\bar{w}^*$ at 70 hPa as delivered for the CCMI models agrees with the derived $\bar{w}^*_{\bar{v}*}$ within <10% for 4 models, while in 3 models to difference is close to or above 20%. After consulting the modeling groups it turned out that the three models with the largest difference (EMAC, NIWA-UKCA, SOCOL) indeed used the actual density to calculate $\bar{w}^*$ (i.e. Equ. 1) instead of the fixed scale height.

5  This inconsistent calculation method inhibits quantitative model comparison of the delivered $\bar{w}^*$ fields, as differences due to different methods are of similar magnitude than the model differences (as can be seen from Fig. 2).

To ensure consistent treatment of model data, we used in our study the $\bar{w}^*$ fields derived from $\bar{v}^*$. While this calculation also introduces errors due to the vertical integration and meridional derivation, it is ensured that the model data is treated in the same way for all models. The error made in the derivation of $\bar{w}^*_{\bar{v}*}$ is shown for the EMAC model in the right panel of Fig. 1:

10  the difference between the $\bar{w}^*$ calculated directly from $\bar{\omega}*$ with the scale height and the $\bar{w}^*_{\bar{v}*}$ derived from $\bar{v}^*$ (blue line) is smaller 5%, i.e. much smaller than the difference due to scale height versus density transformation.

Based on those results we strongly encourage anyone working with residual velocities from multi-model comparison projects to:

1. Check if the given residual vertical velocities $\bar{w}^*$ are consistently calculated by comparison to the derived $\bar{w}^*_{\bar{v}*}$ from $\bar{v}^*$

[Figure]

**Figure 2.** Climatological mean $\bar{w}^*$ over 20°N-20°S at 70 hPa for all CCMI REF-C1 model simulations used in this study. The left bar is the $\bar{w}^*$ delivered by the models, the right bar is $\bar{w}^*_{\bar{v}*}$ derived from $\bar{v}^*$. The number at the top is the percentage difference.

2. If necessary, use those derived $\bar{w}^*_{\bar{v}*}$ values for quantitative model comparison

---

## Author Comment (AC3) · 12 Apr 2018

**Reply to Referee E. Ray (ACP-2017-1143)**

We thank E. Ray for the positive and constructive comments on our manuscript. Below we summarize our answers to his specific comments. Moreover the manuscript is changed taking into account the comments (changed manuscript with changes highlighted is attached to this reply) .

Note that during the review process we found, that the calculation of w* is treated inconsistent within the different models, as in some models a fixed scale height was used to transform w* from Pa/s to m/s, while in other models the actual density was used for this transformation. The different calculation methods of w* can lead to large differences in w*. To facilitate a quantitative model comparison we recalculated w* from the given v* fields. Thus we recalculated the RCTTs, as well as the mixing efficiencies, however the conclusions of the paper do not change. We now provide a supplement (attached to this reply).

**Specific issues:**
p 2, line 11: switch "also" and "be"
→ *done*

p7., line 13: "Karlsruhe, Germany" seems to be a mistake
→ *Thank you, error fixed.*

Pg. 8, line 30: remove "allows" and change to "separates"
→ *done*

Pg. 10 and references: Problem with the "Ray and Andrews, 2017" reference.
→ *Thanks, corrected.*

Pg. 11, line 22: add "of" after "question"
→ *done*

Pg. 16: I was unsure in reading the first paragraph of Section 4.2 whether the correlations shown in Figure 4 were done for all of the models or not. At the end of the paragraph you do mention the sample size of 17 models but I would mention that up front to make it clearer. How much variability in the correlations is there between models?
→ *Thank you for pointing this out, we will move the information about the 17 models up in the paragraph. In fact, the correlation seems to be quite robust. Excluding the one or*

*the other model from this analysis hardly changes the overall picture. We will add some words on this, however, we did (and still do) not see the need for a decent analysis here, so we will keep it very basic.*

Pg. 16, lines 22,23: add commas after "RCTTs" and "tropical pipe"
→ *done*

Pg. 17, line 16: add "a" before "measure"
→ *done*

Pg. 18, line 10: change to "different" and remove "strong relative"
→ *rephased: "mixing leads to different magnitudes in the relative enhancement of AoA."*

Pg. 18, line 20: remove "exemplary"
→ *done*

Pg. 22, line 5: remove "also" and add comma in that space, change "in parts" to something like "partly"
→ *done*

Pg. 24, line 3: change "to" to "by"
→ *done*

Pg. 24, line 6: remove "here"
→ *done*

Pg. 24, lines 8-9: ". . .twice in the figure, once each for the CCMI-1 and CCMVal-2 simulations."
→ *done*

Pg. 24, figure 8: A more descriptive title on the figure would be helpful.
→ *We will switch title and x-Axis description and change the caption as follows: ... contribution on tropical upwelling (calculated as EPFD contribution of downward control calculated tropical upwelling divided by overall tropical upwelling) 30 ...*

Pg. 25, lines 2-3: The statement that wave driving differences can't explain mixing efficiency differences might be too strong based on the small sample size statement made later in the paragraph.
→ *We will extend the sentence by:*
*...has to be rejected for now, however, more model data is required to ultimately confirm this statement.*

Pg. 25, line 9: change to "explain"
→ *changed*

Pg. 25, line 12: change to "influence"
→ *changed*

Pg. 25, line 31: change to "sample"
→ *changed*

Pg. 25-26: The discussion of the model's advection schemes and resolution in these two pages could be shortened in my opinion. The results are interesting but the discussion section is long. Since there is no systematic relationship found among the advection schemes it would be sufficient to just mention that without going into the details.
→ *We tried to reword the section, in order to shorten it. However we think all information is important, thus we still like to include them.*

Pg. 27, line 23: remove "could" and change to "showed"
→*done*

Pg. 27, line 25: add comma after "Thus"
→ *done*

Pg. 28, line 3: change "is" to "were"
→*changed*

Pg. 28, line 4: change "do" to "did", change "also do except" to "expect"
→ *changed*

Pg. 28, line 6: remove "does" and change to "caused"
→ *done*

Pg. 29, line 6: change to "explain"
→ *changed*

[revised manuscript text omitted]

| GEOSCCM | ? | 2.0°x2.5°, L72 | 0.015hPa | FFSL |
| LMDZrepro | ? | 2.5°x3.8°, L50 | 0.07hPa | FV |
| MRI | ? | T42, L68 | 0.01hPa | STFD* |
| SOCOL | ? | T30, L39 | 0.01 hPa | SL* |
| ULAQ | ? | 11.5°x22.5°, L26 | 0.04 hPa | FFEE |
| UMUKCA-METO | ? | 2.5°x3.8°, L60 | 84km | SL |
| WACCM | ? | 1.9°x2.5°, L66 | 0.00005hPa | FFSL |
| **CCMI-1** | | | | |
| ACCESS-CCM | ??? | 2.5°x3.8°, L60 | 84km | SL |
| CMAM | ? | T47, L71 | 0.0008hPa | SP |
| | ? | | | |
| CESM1-WACCM | ?? | 1.9°x2.5°, L66 | 140kCCMI$_2$0171203.texm | FFSL |
| | ? | | | |
| EMAC-L90 | ?? | T42, L90MA | 0.01hPa | FFSL |
| EMAC-L47 | ?? | T42, L47 | 0.01hPa | FFSL |
| GEOSCCM | ?? | 2°x2.5°, L72 | 0.015hPa | FFSL |
| | ?? | | | |
| MRI | ? | TL159, L80 | 0.01 hPa | SL |
| | ?? | | | |
| SOCOL | ?? | T42, L39 | 0.01 hPa | FFSL |
| NIWA-UKCA | ?? | 2.5°x3.8°, L60 | 84km | SL |
| | ? | | | |

[revised manuscript text omitted]
-1 include data requests for the residual velocities $\bar{v}^*$ and $\bar{w}^*$. The $\bar{w}^*$ fields are specified to be given in units of $m/s$. However, as most models use hybrid-pressure coordinates, the vertical velocity $\bar{\omega}^*$ in $Pa/s$ is usually available and has to be transformed to $m/s$. The exact transformation of $\omega$ to w is given by:

$$\quad \omega \approx \frac{dp}{dt} = \frac{\partial z}{\partial t}\frac{\partial p}{\partial z} = \text{w}\frac{-pg}{RT} \tag{1}$$

I.e. on a given pressure level $p$ the transformation is dependent on the local temperature $T$. Alternatively, a constant scale height assumption can be made for the transformation:

$$\omega \approx \text{w}\frac{-p}{H} \tag{2}$$

In the CCMI data request a scale height of H = 6950 m was suggested. Fig. 1 compares the profile of tropical mean $\bar{w}^*$
10   from one model (EMAC) calculated with the actual density (Equ. 1, red line) and with the scale height (Equ. 2, black line). Differences in the lower stratosphere (between 100 to 70 hP) are more than 20%. Upon close inspection of delivered CCMI model output, it turned out that some model data was processed converting vertical velocities with the actual density while others used a fixed scale height. Given the large deviations of the two methods, this inconsistency has to be taken into account for a quantitative model comparison. In the following we give an outline how those inconsistencies can be detected, and how a
15   consistent model comparison can be achieved.

Given continuity, the residual vertical velocity is linked to the meridional velocity:

$$\frac{\partial \bar{v}^*}{\partial y} + \frac{\partial \bar{\omega}^*}{\partial p} = 0 \tag{3}$$

Therefore, it is possible to deduce the residual vertical velocity $\bar{\omega}^*$ from the residual meridional velocity by vertical integration and meridional derivation. This vertical velocity $\bar{\omega}^*$ can then again be converted to $\bar{w}^*$ using the given scale height. By
20   comparing the given $\bar{w}^*$ from the model output to the $\bar{w}^*$ that is calculated from $\bar{v}^*$ (referred to as $\bar{w}^*_{\bar{v}^*}$ in the following) it can

[Figure]

**Figure 1.** Profile of climatological mean $\bar{w}^*$ over 20N-20S for EMAC-L90. Left: $\bar{w}^*_{CCMI}$ delivered to CCMI with actual density used in the transformation (red), $\bar{w}^*_H$ with scale height used for the transformation (black) and $\bar{w}^*_{\bar{v}*}$ derived from $\bar{v}*$ (blue). Right: relative difference between $\bar{w}^*_{CCMI}$ to $\bar{w}^*_H$ (red) and relative difference between $\bar{w}^*_{\bar{v}*}$ and $\bar{w}^*_H$ (blue).

be tested whether the same scale height was used originally in the calculation of $\bar{w}^*$. As shown in Fig. 2, tropical mean $\bar{w}^*$ at 70 hPa as delivered for the CCMI models agrees with the derived $\bar{w}^*_{\bar{v}*}$ within <10% for 4 models, while in 3 models to difference is close to or above 20%. After consulting the modeling groups it turned out that the three models with the largest difference (EMAC, NIWA-UKCA, SOCOL) indeed used the actual density to calculate $\bar{w}^*$ (i.e. Equ. 1) instead of the fixed scale height.

5    This inconsistent calculation method inhibits quantitative model comparison of the delivered $\bar{w}^*$ fields, as differences due to different methods are of similar magnitude than the model differences (as can be seen from Fig. 2).

To ensure consistent treatment of model data, we used in our study the $\bar{w}^*$ fields derived from $\bar{v}^*$. While this calculation also introduces errors due to the vertical integration and meridional derivation, it is ensured that the model data is treated in the same way for all models. The error made in the derivation of $\bar{w}^*_{\bar{v}*}$ is shown for the EMAC model in the right panel of Fig. 1:

10   the difference between the $\bar{w}^*$ calculated directly from $\bar{\omega}*$ with the scale height and the $\bar{w}^*_{\bar{v}*}$ derived from $\bar{v}^*$ (blue line) is smaller 5%, i.e. much smaller than the difference due to scale height versus density transformation.

Based on those results we strongly encourage anyone working with residual velocities from multi-model comparison projects to:

1. Check if the given residual vertical velocities $\bar{w}^*$ are consistently calculated by comparison to the derived $\bar{w}^*_{\bar{v}*}$ from $\bar{v}^*$

[Figure]

**Figure 2.** Climatological mean $\bar{w}^*$ over 20°N-20°S at 70 hPa for all CCMI REF-C1 model simulations used in this study. The left bar is the $\bar{w}^*$ delivered by the models, the right bar is $\bar{w}^*_{\bar{v}^*}$ derived from $\bar{v}^*$. The number at the top is the percentage difference.

2. If necessary, use those derived $\bar{w}^*_{\bar{v}^*}$ values for quantitative model comparison